# Review article: Fire emissions in the Brazilian Cerrado: dynamics, estimates, management and their role in the global carbon budget

Renata Moura da Veiga[1], Celso von Randow[1], Chantelle Burton[2], Douglas I. Kelley[3], Manoel Cardoso[1], Fabiano Morelli[1]

[1]National Institute for Space Research (INPE), Avenida dos Astronautas 1758, 12227-010, Sao Jose dos Campos, SP, Brazil
[2]Met Office Hadley Centre, FitzRoy Road, Exeter, Devon, EX1 3PB, United Kingdom
[3]UK Centre for Ecology and Hydrology, Maclean Building, Bensor Lane, Crowmarsh Gifford, Wallingford, Oxfordshire, OX10 8BB, United Kingdom

*Correspondence to*: Renata Moura da Veiga (rmvrenata@gmail.com)

**Abstract.** Estimating fire emissions in the Brazilian Cerrado requires integrating fire parameters, mitigation strategies and policies. Despite the Cerrado's significant contribution to global fire emissions, research in this area is still overlooked when compared to other savanna ecosystems. Here, we provide a comprehensive understanding of the Cerrado's fire emissions within the global carbon budget by examining how fire dynamics, management and policy shape emissions. We systematically reviewed 77 papers, of which 57% address fire dynamics, management and policy. While these are key to providing a holistic understanding of fire emissions, linking them to estimates is challenging, especially due to the difficulty in valuing the qualitative aspects of fire. This review only identified two papers that explicitly analyze fire emissions in the Cerrado, and found that 17% of papers are led by institutions located within the Cerrado biome area. These numbers reinforce the urgent need for further investigation into the topic. Most papers employ different methods to achieve their results. Evidence suggests growing interest in fire emissions in the Cerrado, reflected in the rising number of studies over the years. More research is required to provide a more comprehensive understanding of fire emissions in the Cerrado, understand fire dynamics and emissions, and identify potential mitigation measures that could help reduce the Cerrado's contribution to the global carbon budget. This could be achieved by better accounting of emission parameters across the Cerrado's vegetation types and fire regimes, and by including fire management representation in land surface models and using observational data to constrain and assess their utility.

## 1 Introduction

The Brazilian Cerrado biome (hereafter referred to as "the Cerrado") covers approximately 2 million $km^2$ in Central Brazil. It is the second largest biome in Brazil, covering almost 24% of the country (IBGE, 2004). Around 49% (965,783 $km^2$) of the Cerrado is covered by natural vegetation (MapBiomas, 2022), where anthropogenic land uses do not occur (e.g. pasture and agriculture). The Cerrado is one of the species-rich tropical savannas in the world (Klink and Machado, 2005; Schmidt and Eloy, 2020; Simon et al., 2009), and it is heavily influenced by fire (Fidelis, 2020; Pivello, 2011). Fire is the most critical factor in maintaining its health and biodiversity (Fidelis, 2020; Franke et al., 2018; Pivello, 2011), and the wide variety of ecosystems contained within it (Franke et al., 2018). Fire is also a major driver of carbon emissions, which influences the Cerrado's overall contribution to the global carbon budget.

In the Cerrado, the extensive range of vegetation assemblages influences fire through variations in fuel type and structure across the region, as well as microclimate conditions (Flannigan et al., 2009; Gomes et al., 2020a). The Cerrado

comprises a mosaic of vegetation formations, or physiognomies, that range from grasslands (such as *Campo Limpo* and *Campo Sujo*) to savannas (such as *Cerrado Ralo*, *Cerrado Típico*, and *Cerrado Denso*) to forest formations (such as *Cerradão*) (Ribeiro and Walter, 2008). The open formations (grasslands and savannas) are dominated by grasses and herbs, or fine fuel load, with few shrubs and trees (Ribeiro and Walter, 2008). Open formations have higher wind speed, higher air temperature, lower relative humidity, and lower fuel moisture than forests (Hoffmann et al., 2012).


The Cerrado presents two distinct seasons: the rainy season (November–March), characterized by biomass accumulation and higher fuel moisture, and the dry season (April–October), when the accumulated biomass available for burning becomes highly flammable, and fire can rapidly spread (Gomes et al., 2020b; Hoffmann et al., 2012; Silva et al., 2021). Intense wildfires in the Cerrado tend to occur in the late dry season (LDS; August–October) (Moura et al., 2019; Silva et al., 2021), when fire emissions are expected to be higher.


In the Cerrado, higher temperatures and reduced precipitation are now more common due to climate change (Hofman et al., 2021), which also change its fire regimes. Arruda et al. (2024) observed a shift in the peak of fire activity, with the highest concentration of fire events changing from September to October between 1985 and 2022. Fire events are also becoming more frequent and intense (Gomes et al., 2024; Oliveira et al., 2022; Pivello et al., 2021). Working groups I and II of the Sixth Assessment Report of the Intergovernmental Panel on Climate Change (IPCC AR6 WGI/WGII; IPCC, 2021, 2022) and the United Nations Environment Programme "Spreading like Wildfire" report (UNEP, 2022) warn that climate change increases drought conditions, which can aggravate heatwaves, increasing the risk of fire occurrence and the intensity and frequency of extreme events, such as wildfires. This happens because the combination of extreme weather events that occur simultaneously, or compound events, can amplify their effects (dos Santos et al., 2024). Drought-heatwaves episodes and extreme fire events intensified by climate change also impact hydrological processes, including precipitation and evaporation trends, groundwater recharge and soil infiltration capacity (Klink et al., 2020; Libonati et al., 2022). This is particularly important because the Cerrado region supports aquifers that supply major hydrographic basins in the whole country (Klink et al., 2020).



Fires in the Cerrado release substantial amounts of carbon into the atmosphere, primarily in the form of carbon dioxide ($CO_2$), but also as other greenhouse gases such as carbon monoxide (CO) and methane ($CH_4$) – $CO_2$ and CO combined account for 95% of the carbon emitted during biomass burning (Ward and Hardy, 1991). $CO_2$ and CO are both involved in atmospheric chemistry and in the greenhouse effect in different ways. Savanna burning dominates the emission of CO through incomplete combustion due to limited oxygen (Ehhalt et al., 2001). Similarly, $CO_2$ is released during complete combustion of biomass burning (Prentice et al., 2001). $CO_2$ is a major greenhouse gas, and it is crucial in absorbing and trapping infrared radiation in the atmosphere. The increased concentration of $CO_2$ in the atmosphere has warmed the Earth in alarming amounts (IPCC, 2022). Thus, understanding the contribution of each of these gases, especially $CO_2$, in fire emissions is essential, particularly in fire-prone settings such as the Cerrado.



Beyond emissions, fire interacts with several components of the carbon cycle, shaping complex processes over time (Bond et al., 2004). For example, post-fire recovery critically shapes the Cerrado's long-term carbon balance (Gomes et al.,

2020b; 2024; Hamilton et al., 2024). If vegetation fully regenerates to its pre-fire state, there is no net effect on atmospheric $CO_2$ levels over time. Alternatively, if fire activity decreases and vegetation accumulate, the landscape may shift to a net carbon sink. Conversely, if fires reduce long-term vegetation cover, the Cerrado could become a sustained carbon source, as observed globally (Burton et al., 2024; Jones et al. 2024). In fact, Gomes et al. (2024) indicate that fires in the Cerrado have acted as a source of carbon emissions to the atmosphere over the past decades.

The Cerrado's fires are potentially responsible for more than 30% (about 0.13 PgC year[-1]) of Brazil's total fire emissions (da Silva Junior et al., 2020). The Cerrado also accounts for about 14% of Brazil's emission from land use and land cover change (SEEG, 2023). Brazil is the highest emitter in the world in this category (Friedlingstein et al., 2023), contributing with up to 0.4 PgC year[-1] (Rosan et al., 2021). Thus, understanding the Cerrado's fire-driven carbon emissions is critical for accurately quantifying national greenhouse gas contributions, assessing Brazil's progress toward climate commitments, such

as its Nationally Determined Contributions (NDCs), and evaluating its role in the global carbon budget. Accurate quantification of these emissions helps evaluate carbon sources and sinks, greenhouse gas contributions, and the impact of land management and climate policies at both national and global scales.

        In this context, understanding fire dynamics provides grounding for assessing fire emissions in the Cerrado, and the interaction between these is essential for uncovering the factors that influence the Cerrado's role in the global carbon budget

and the broader implications for national and international policy. Linking fire dynamics to estimated emissions also guides mitigation by identifying aspects for potential intervention. For example, natural activities to avoid the release of greenhouse gases (GHG) or to increase carbon storage by letting vegetation recover, termed natural climate solutions (NCS), have emerged as a possibility to reduce emissions (Griscom et al., 2020). Tropical NCS, like improved fire management, could mitigate 6.56 petagrams of $CO_2$ annually (Pg $CO_2$ year[-1]) between 2030 and 2050 (Griscom et al., 2020). Management activities, which

include fire management, hold 26% of NCS' mitigation potential in Asia, Africa and Latin America (Griscom et al., 2020). Lipsett-Moore et al. (2018) found that prescribed savanna burning, a typical fire management activity, could reduce 75% of emissions from LDS fires in South America.

        Thus, it is essential to account for fire emissions and recognize mitigation mechanisms worldwide. This is notably relevant for the Cerrado due to its intrinsic connection to fire. This review examines how fire carbon emissions in the Cerrado

are estimated, particularly what are the parameters used to estimate emissions, what fire carbon emissions estimates have been published, and their implications on fire management and policy. Since carbon is a major contributor to atmospheric greenhouse gas levels, this systematic review focuses on emissions in terms of carbon released by fire, including all the carbon components emitted during biomass burning, or in terms of $CO_2$ alone, due to its impact on the greenhouse effect.

        This systematic literature review aims to gain a comprehensive understanding of the Cerrado's fire emissions within

the global carbon budget by evaluating how fire parameters can inform emission estimates and mitigation strategies. Particularly, we aim to: (a) outline current emissions estimates, specially carbon, in regions that encompass the Cerrado or are limited to it; (b) describe fire parameters that support these estimates; (c) understand how these estimates fit the carbon budget; (d) identify mitigation strategies in the biome and their link to fire-driven carbon fluxes; and (e) identify research gaps. This

will support improvements for future fire emission estimation, provide insights into the placement of the Cerrado in the global carbon balance and assist fire policies.

## 2 Methods

According to Moher et al. (2009, 1), a systematic review is "*a review of a clearly formulated question that uses systematic and explicit methods to identify, select, and critically appraise relevant research, and to collect and analyze data from the studies that are included in the review*". It is based on rigorous criteria and a well-established and reproducible methodological approach to evaluate and synthesize the state of understanding of a specific topic (Cronin et al., 2008; Foo et al., 2021; Moher et al., 2015). We conducted this systematic review according to the Preferred Reporting Items for Systematic reviews and Meta-Analyses (PRISMA) Statement (Moher et al., 2009), and the practical guide on conducting systematic searches by Foo et al. (2021). PRISMA was designed to guide authors in transparently reporting on the review process (Moher et al., 2009).

We followed the four main steps outlined by Foo et al. (2021) to conduct this systematic review: (1) decide on a review question; (2) execute search; (3) initial literature screening; and (4) full-text screening. We also combined the literature review steps outlined by Cronin et al. (2008), which includes selecting a review topic, searching the literature, gathering, reading, analyzing the literature, and writing the review. After establishing our research question as "How compiling published material on fire emissions in the Cerrado can provide a holistic understanding of their role in the global carbon budget?", we combined keywords in English with the Boolean operators AND and OR in the Google Scholar database: fire AND emissions AND (Cerrado OR "Brazilian Savanna" OR (Savanna AND Brazil)). Keyword searches revealed a total of 3,717 records.

We applied four inclusionary criteria to identify relevant literature: papers had to be (1) published in peer-reviewed journals with an impact factor greater than 1; (2) encompass the Cerrado biome; (3) be published after 2003; and (4) be conducted in areas that do not explicitly include anthropogenic land uses. Although we acknowledge the role of anthropogenic fires and the importance of further research to integrate these to fully assess fire emissions in the Cerrado, we focus on fires that are not explicitly used for anthropogenic land uses – as land clearing for agriculture implementation – to provide a clearer ecological perspective on fire emissions in the Cerrado and their implications for the global carbon budget. Thus, identifying the key drivers of fire emissions in the Cerrado's landscapes provide a strong basis for improving emissions estimates, understanding fire-climate feedback, and assessing long-term ecosystem resilience in the Cerrado.

To improve the assessment of our research question, we have also incorporated in our review papers that don't focus only on the Cerrado but rather include it as part of the analysis. Although we explicitly limited the region to the Cerrado in our keyword search, we often encountered papers that include the Cerrado but are not limited to it. Including these papers in our review allows for a complete analysis of fire emissions in the Cerrado, from global to local scales.

Despite its constraints, the impact factor was included as an indicator of scientific quality (Andersen et al., 2006; Ketcham and Crawford, 2007). The search starts in 2003 because fire emissions estimation is highly dependent on satellite

data, much of it from MODIS (Moderate-Resolution Imaging Spectroradiometer) satellite burn products, with full-year data available starting in 2003. The influence of MODIS data on our understanding of global fires can be seen in fire-vegetation modelling. There is broader disagreement between models on simulated burned area before the MODIS era, and much higher agreement during the satellite period (Kloster and Lasslop, 2017; Hantson et al., 2020; Rabin et al., 2017). We evaluate the trend in the number of papers published over time using linear regression.

The criteria led to the initial screening of 109 papers. Although we used keywords to conduct our review, the searches still returned papers not in English, or that did not mention fire emissions. 32 papers were excluded due to being duplicates, not in English, or not mentioning fire emissions. Review and perspective papers were included in this systematic literature review to contribute to a more complete analysis of fire emissions in the Cerrado. Review and perspective papers analyze previously published studies by evaluating existing literature (review) or expressing opinions on a specific topic (perspective) while empirical studies provide new information based on observation or experiments. Although they do not focus on bringing original research, they supply the current knowledge of a specific topic and highlight pertinent published literature (Cronin et al., 2008). We full-text screened the remaining 77 papers to confirm they met all the eligibility criteria. Figure 1 demonstrates the systematic literature review process through the PRISMA diagram.

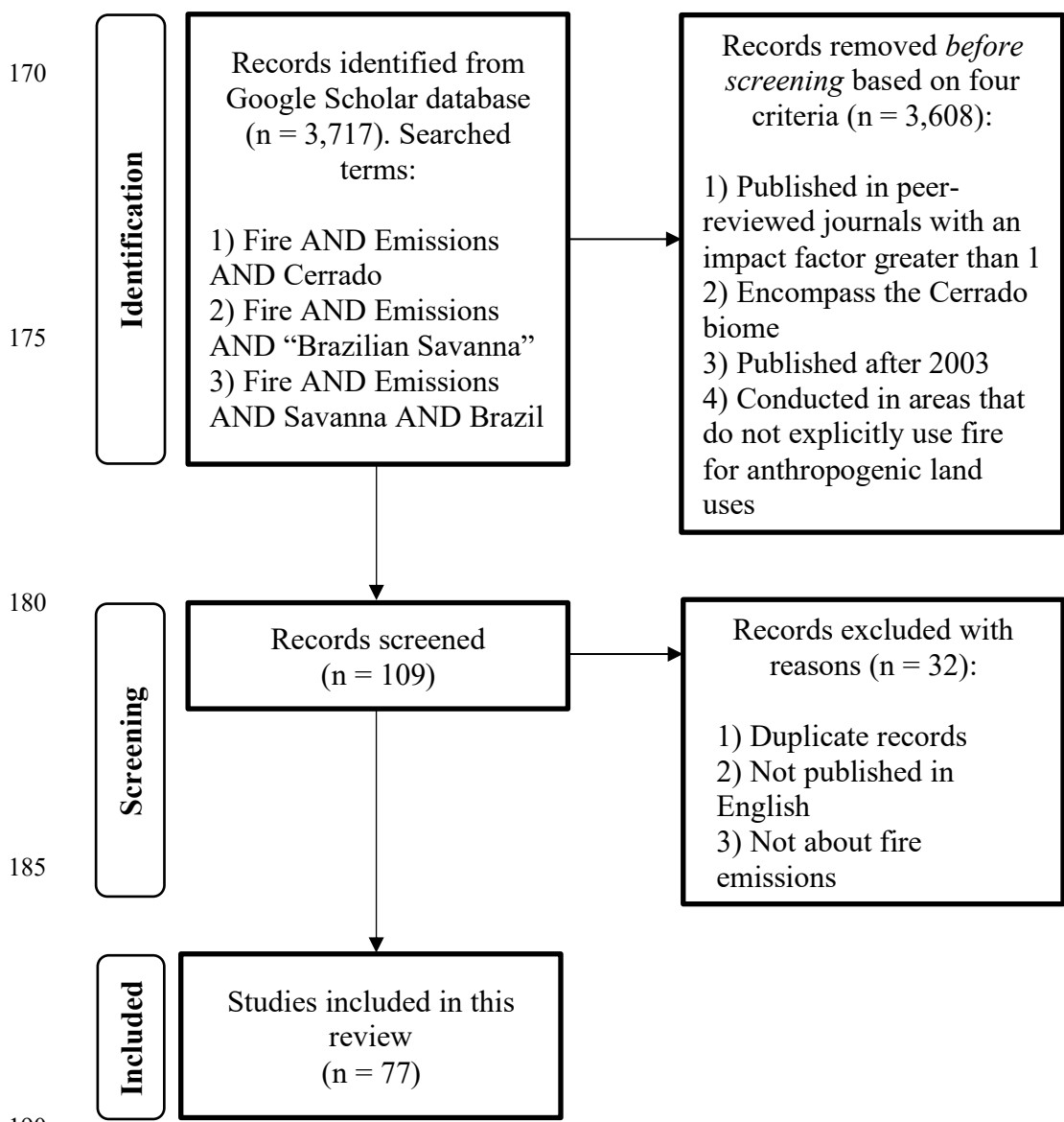

**Identification of studies via Google Scholar database**

**Identification**

Records identified from Google Scholar database (n = 3,717). Searched terms:

1) Fire AND Emissions AND Cerrado
2) Fire AND Emissions AND "Brazilian Savanna"
3) Fire AND Emissions AND Savanna AND Brazil

Records removed *before screening* based on four criteria (n = 3,608):

1) Published in peer-reviewed journals with an impact factor greater than 1
2) Encompass the Cerrado biome
3) Published after 2003
4) Conducted in areas that do not explicitly use fire for anthropogenic land uses

**Screening**

Records screened (n = 109)

Records excluded with reasons (n = 32):

1) Duplicate records
2) Not published in English
3) Not about fire emissions

**Included**

Studies included in this review (n = 77)

**Figure 1: Adapted PRISMA flow diagram demonstrating the systematic literature review process divided into three steps: identification of potential papers through searched terms in the Google Scholar database, and exclusion of papers based on the four criteria established for this research; screening of the papers selected and exclusion of papers with the reported reasons; and inclusion of papers in this literature review.**

Once we had selected the papers, we adopted a systematic scheme to review, analyze, and synthesize them. We retrieved the following information from the 77 papers: title, year of publication, authors, journal, study design, area of study,

what is measured, how it is measured, and institutions of authors. The 77 papers selected for this literature review are available in the Supplement. After selecting the papers, we wrote a summary of each study to simplify the understanding of its content and build a solid basis for writing the review (Cronin et al., 2008). From this, we framed the review by dividing the literature into themes. This approach has proven to be an efficient way to conduct and discuss the results from a systematic literature review in da Veiga and Nikolakis (2022).

We classified the reviewed papers based on (a) location range, from global to local scales: global, tropical region, South America, Brazil and Cerrado; (b) topic of research, where we identified three topics in the literature: fire dynamics parameters, fire emission estimates, and fire management and policy; and (c) study design, divided into empirical, review, and perspective. These classifications allowed a deep understanding of the general scope of published literature on fire emissions in the Cerrado, including the purpose of the studies. This, in turn, enabled the identification of key trends for future research, which are outlined in this review. We divide our results into four sections, first summarizing the overall trends in current publications, and then highlighting the key findings from each of the three literature topics identified through our review process.

## 3 Results

### 3.1 Systematic literature review process

We reviewed 77 papers and incorporated them into the literature review process. The number of papers published has increased since 2003 (Fig. 2), indicating a rise in interest in understanding fire emissions in the Cerrado and a rise in papers on fire science being published.

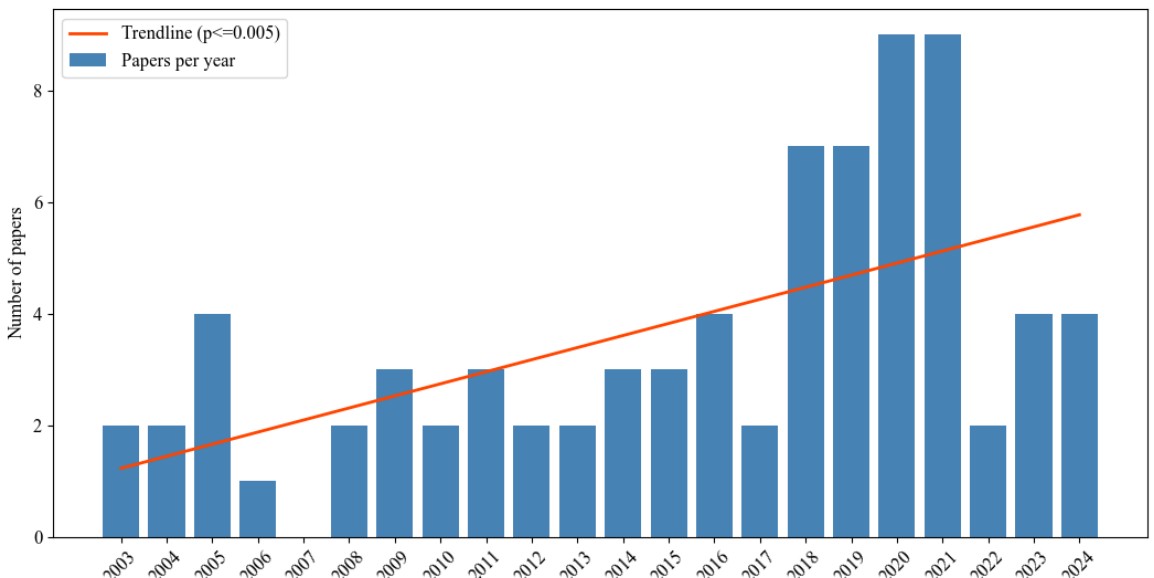

**Figure 2: Number of papers published per year, since 2003, included in the literature review (p-value ≤ 0.005).**

There is a statistically significant increasing trend ($p \leq 0.005$) in the number of papers published throughout the time series, with a sharp drop in publications in the year 2022. 28 papers focused on the Cerrado, while 25 provide a global scale analysis, 9 relate to Brazil, 9 to South America, and 6 to the Tropical region. These numbers suggest that most papers are either restricted to the Cerrado, or provide a broad global analysis, with fewer papers in between, as shown in Fig. 3.

Papers not restricted to the Cerrado provide a broader perspective on emissions, encompassing the biome instead of
235 being limited to it. Often papers referred to one limited region were expanded to represent a broader area. For example, within the South American Biomass Burning Analysis (SAMBBA) Project in 2012, Hodgson et al. (2018) used airborne flights in Tocantins, a Brazilian state dominated by the Cerrado, to represent fire emissions from smoke sampling for the whole Cerrado biome. Similarly, Mistry et al. (2019) use the example of Brazil and Venezuela to illustrate how fire management can support emission reduction in South America.

As per the study design, we identified 60 empirical papers (78%), 14 review papers (18%), and 3 perspective papers (4%) We further connect the study design with the coverage of the study area in Fig. 3.

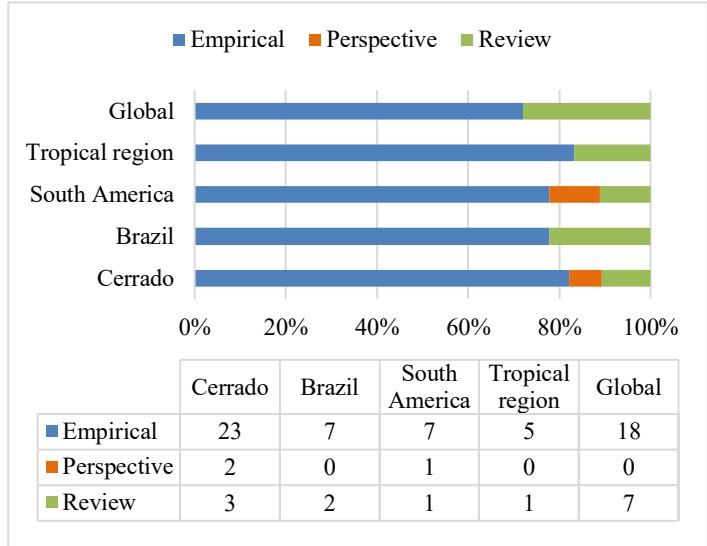

| | Cerrado | Brazil | South America | Tropical region | Global |
|---|---|---|---|---|---|
| ■Empirical | 23 | 7 | 7 | 5 | 18 |
| ■Perspective | 2 | 0 | 1 | 0 | 0 |
| ■Review | 3 | 2 | 1 | 1 | 7 |

**Figure 3: Number of papers per study design and per coverage of study area in both percentage (chart) and absolute numbers (data table).**

We also observed that international (non-Brazilian) institutions drive most of the research captured by this literature review. We gathered the institution from the first author of each paper, of which 47 are international (61%) and 30 are Brazilian (39%). From the Brazilian-led papers, 13 (43.3%) are from institutions located within the Cerrado biome area (Fig. 4). From
260 the international-led studies, 16 papers (34%) involve authors from Brazilian institutions, while 14 of the Brazilian-led studies (47%) include authors from international institutions.

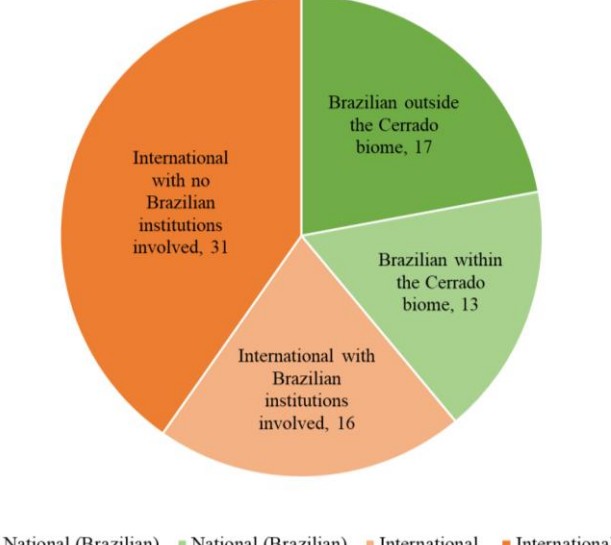

■ National (Brazilian) ■ National (Brazilian) ■ International ■ International

**Figure 4: Institutions of the first authors from the papers reviewed. 47 papers involve first authors from international (non-Brazilian) institutions (Oranges), while 30 come from Brazilian institutions (Greens). From the international-led papers, 16 involve authors from Brazilian institutions, while 31 do not. From the Brazilian-led papers, 13 are from institutions located within the Cerrado biome, while 17 are not.**

By reading the papers and through full-text screening of the content, we identified three topics of search from this systematic literature review process: (1) fire dynamics parameters, where the study aims to measure and evaluate parameters that can further be used to estimate emissions; (2) emission estimates, which include papers that focus on the quantification of fire emissions; (3) fire management and policy, where papers discuss the importance of fire management and fire policy to influence emission rates.

Of the 77 papers reviewed, 46 relate to fire dynamics parameters used to estimate emissions, 43 report the amounts of fire emissions, and 12 report fire management and policy. It's worth noting that 24 papers are related to more than one topic. These numbers indicate that most papers are not related to reporting emissions but provide information to support the understanding and estimation of fire emissions – 57% (double counts included) of papers address fire dynamics, management and policy.

Of the 43 papers that report fire emissions, 23 focus only on fire emissions, while 18 analyze fire emissions and fire dynamics parameters, 1 focuses on fire emissions and fire management, and 1 on the three topics. From the 23 papers exclusive to fire emissions, only 2 are restricted to the Cerrado, one focusing on net $CO_2$ (Gomes et al., 2024), and the other on fine particulate matter (Mataveli et al., 2019). The remaining papers include the Cerrado region but are not limited to it (9 provide a global analysis, 5 relate to the Tropical region, 4 to South America and 3 to Brazil).

Most papers provide new information on the parameters used to estimate fire emissions and on emissions themselves rather than analyzing existing literature. These often handle modelling techniques, satellite observations, in situ observations

and even results found in previous studies, reflecting the importance of literature reviews in supporting the development of new data. In this study, we discuss 'models' in terms of the qualitative and quantitative characterizations of components within a system and their interactions (IPBES, 2016). These tools are often combined to provide a more complete analysis. Modelling and satellite observation are frequently integrated, with satellite observations being data providers to models, or models being evaluated by satellite observations, or even through data assimilation between satellite data and modelling simulations. From
our results, 22 empirical papers integrate modelling and satellite data, 4 integrate modelling and literature review, 4 integrate satellite and in situ observations, and 1 integrates modelling, satellite data, literature review and in situ observations. Figure 5 shows the techniques used across study areas in empirical papers.

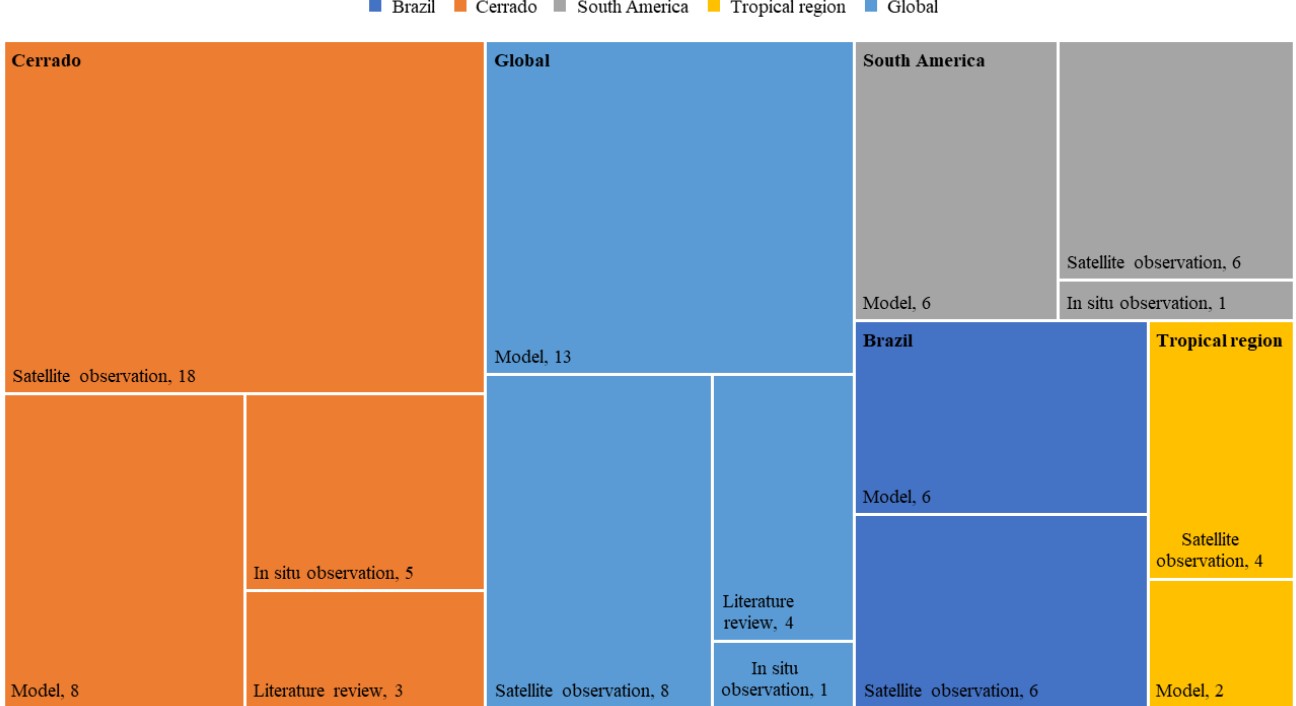

**Figure 5: Treemap of the techniques used across study areas in empirical papers. The numbers represent the number of studies of**
295 **each technique within each study area. The study areas Global, Tropical region, South America and Brazil are regions that include**
**results for the Cerrado. 26 papers combine different techniques and are double counted.**

Of the 30 papers specific to the Cerrado, 4 were classified as review papers, and these are mostly related to the parameters associated with fire emissions, fire behavior, and fire ecology (see Arruda et al., 2018; Bustamante et al., 2012; Gomes et al., 2018; Gomes et al., 2020a). We further synthesize the main findings of the three topics from the systematic
literature review process.

### 3.2 Fire dynamics parameters to estimate fire emissions

Papers under 'fire dynamic parameters' encompass 46% of the studies reviewed, underscoring the importance of variables like burned area, fuel characteristics, combustion completeness, combustion efficiency and emission factor in fire emissions research. These parameters directly influence emission estimates, with their combination playing key roles in determining carbon emissions from fires. By examining these variables within the specific ecological and climatic context of the Cerrado, we gain insights into how fire behavior and emissions in this biome interact.

The prevalence of studies on fire dynamics parameters reflects the accessibility of these variables and indicates the importance of linking fire dynamics directly to emissions, with studies often highlighting the potential applicability of their research in fire emission estimation (e.g. Libonati et al., 2015; Pereira Junior et al., 2014). This focus on fire dynamics provides some of the most current information available, yet it suggests a need for more research to correlate fire drivers to emissions. We further discuss the fire dynamics parameters found in the literature review process.

### 3.2.1 Burned area and fuel characteristics

Burned area detection in the Cerrado has mainly been measured via the satellites Aqua and Terra/MODIS and Landsat/TM (see Libonati et al., 2015; Oliveira et al., 2021; Pereira et al., 2021) and is one of the primary parameters for estimating emissions. The Global Fire Emissions Database version 5 (GFED5) primarily relies on MODIS products to provide global burned area (Chen et al., 2023). GFED5 estimates 21.35 Mha year$^{-1}$ (2001–2020) for burned area in open savannas of the Southern Hemisphere of South America (SHSA), where the Cerrado is located. GFED5 burned area results are more aligned with data from higher-resolution satellite sensors (Chen et al., 2023), indicating enhancement compared to previous versions, and a potential to improve fire emissions estimates derived from GFED5 (Chen et al., 2023).

Libonati et al. (2015) developed a regional algorithm using MODIS data to increase the accuracy of estimations of burned area in the Cerrado, demonstrating that regional products more accurately capture vegetation diversity. Also, to capture the variety of fire dynamics throughout the biome, Silva et al. (2021) map fire characteristics for the 19 ecoregions of the Cerrado, including the patterns and trends of burned area using MODIS data. Results show a great variation of size of burned area in the Cerrado, with large areas detected in the boundaries with other biomes (Silva et al., 2021). Using Aqua and Terra/MODIS data and the algorithm developed by Libonati et al. (2015), the National Institute for Space Research (INPE, in Portuguese) in Brazil has estimated that 16.2% of the Cerrado was burned in 2007 (the equivalent of 329,138km$^2$), a critical year in terms of wildfires in Brazil, and 3.6% in 2009 (74,085km$^2$), a year with low overall rates of burned area in the country.

The extent of area burned is highly connected to the fuel characteristics, including fuel type (vegetation components), moisture (amount of water content), and load, and these are essential to determine the intensity and occurrence of fire, together with climatic conditions (Alvarado et al., 2017; Gomes et al., 2020b). Fuel type also influences the monthly peak in fire of each vegetation type, due to climatic conditions. Grassy material tends to burn first because of its high flammability, with the fire peak from grassland happening in August (Arruda et al., 2024). In comparison, savannas tend to encounter more fires in

September and October (Arruda et al., 2024). These impact the extent of area burned in each vegetation type. From 1985 to 2022, 85% of burned area in the Cerrado affected native vegetation, with savannas accounting for 54%, and grasslands, 11% (Arruda et al., 2024).

Fuel load in the Cerrado is highly connected to the seasonal variation in precipitation (Oliveira et al., 2021). For example, drought was the strongest predictor of burned areas made by Alvarado et al. (2017). Rainfall in the wet season allows fuel to build up, determining the fuel load available for burning. In contrast, rainfall in the dry season determines the moisture content of the accumulated fuel and, thus, the probability of burning (Alvarado et al., 2017). Fuel load increases with vegetation density (Oliveira et al., 2021), but grasslands have more biomass available for burning than forests due to the fine fuel load in each physiognomy. Approximately 95% of the biomass of the herbs and grass layers is available for burning in the Cerrado. In comparison, only 0.01% of the shrubs and tree layers are available due to the high quantity of woody material and low quantity of grassy material (Gomes et al., 2020b).

The proportion of influence between climate and fuel on the resultant burned area and emissions varies and is still being discussed. In a global context, burned area is likely to increase worldwide due to climate change (Burton et al., 2024). In South America, an increase in temperature and decrease in moisture availability is expected to drive an increase in burned area, including in the Cerrado (Burton et al., 2021). Similarly, Silva et al. (2019) predicts increased burned area in the Cerrado under all IPCC Representative Concentration Pathways (RPC) scenarios. The RCP 2.6, the scenario most closely aligned with the United Nation's 1.5 °C target, could lead to an increase in burned area up to 22% by 2050 in the Cerrado.

### 3.2.2 Combustion efficiency, combustion completeness and emission factor

Combustion efficiency, combustion completeness and emission factor have been identified in the literature as important parameters to estimate fire emissions. Combustion completeness refers to the amount of biomass converted to gas, aerosols and particulates during the combustion process and released into the atmosphere (Carvalho Jr. et al., 1998). Similarly, combustion efficiency identifies "the percentage of carbon released during combustion of biomass fuels in the chemical form of carbon dioxide" (Ward and Hardy, 1991, 117–118). Combustion efficiency is often measured by the amount of $CO_2$ emitted divided by the amount of $CO_2$ and CO emissions combined, termed modified combustion efficiency (MCE; see Andreae, 2019; Hodgson et al., 2018; Vernooij et al., 2021). MCE tends to be higher in open savannas (Vernooij et al., 2023). Values above 0.9 tend to characterize fires in a flaming stage, and these are predominant in the Cerrado due to the dry fine fuel that is likely to rapidly burn (Hodgson et al., 2018).

Using airborne sensors, MCE has been reported to be 0.94±0.02 in a flight above Tocantins in 2012 (Hodgson et al., 2018). Vernooij et al. (2021) used a UAV-based approach (uncrewed aerial vehicle) to sample smoke from grassland and savanna formations in 2017–2018, in a Protected Area also in Tocantins (*Estação Ecológica Serra Geral do Tocantins*, EESGT), to evaluate the seasonal burning differences. The authors concluded that LDS fires (after July 1; more intense fires) have slightly higher MCE when compared to early dry season fires (EDS; before July 1; more mild fires): 0.963 and 0.957, respectively. In further studies, Vernooij et al. (2023) discuss that the MCE found in Vernooij et al. (2021) from measurements

in EESGT may have underestimated the MCE in other parts of the Cerrado when analyzing it in a biome-scale. These values are consistent with other savannas in the world – global MCE for savannas and grasslands averages 0.94 (Andreae, 2019), while MCE in the African and in the Australian savannas have been reported as 0.938±0.019 and 0.86–0.99, respectively (Hodgson et al., 2018).

The emission factor (EF) is an essential component that contributes to estimating emission, and it refers to the ratio of a particular gas released divided by the fuel consumed, expressed in grams of that specific gas per kilogram of dry matter (Palacios-Orueta et al., 2005). EF depends on the type and moisture content of the fuel consumed, amount of fuel burned, amount and concentration of the emitted gas, and meteorological conditions during the fire (Palacios-Orueta et al., 2005; Vernooij et al., 2023). Thus, EF is highly variable across studies and often the source of uncertainty in fire emission estimates. In fact, Andreae (2019) argues that the emission factors of $CO_2$ and CO from forest fires are the main source of uncertainty in relation to the impact of vegetation fires on the global carbon cycle.

Andreae (2019) and Hodgson et al. (2018) argue that local values of EF can be used to improve estimates. Similarly, Vernooij et al. (2023) proposes incorporating dynamic EF to improve representation of savanna's temporal and spatial variabilities, better capturing its fire regime and resultant emissions. When comparing EDS and LDS fires in savannas worldwide, including the Cerrado, Vernooij et al. (2023) finds that fuel moisture content and relative humidity were lower in the LDS compared to the EDS, increasing fuel consumption and fire intensity over the dry season, and resulting in higher EFs in the LDS. The values of EF found from recent studies are summarized in Table 1.

**Table 1: Emission factors values from areas that include the Cerrado found by this review.**

| Study | Study area | Emission Factor | Value (g kg$^{-1}$) |
|---|---|---|---|
| Van Der Werf et al. (2017) | Global – savannas | $EF_{CO_2}$ | 1,686 |
| | | $EF_{CO}$ | 63 |
| | | $EF_{CH_4}$ | 1.94 |
| Andreae (2019) | Global – savannas | $EF_{CO_2}$ | 1,660 |
| | | $EF_{CO}$ | 69 |
| | | $EF_{CH_4}$ | 2.7 |
| Vernooij et al. (2023) | Global – savannas | $EF_{CO_2}$ | 1,685 |
| | | $EF_{CO}$ | 64 |
| | | $EF_{CH_4}$ | 1.85 |
| Hodgson et al. (2018) | Cerrado | $EF_{CO_2}$ | 1,711 ± 175 |
| | | $EF_{CO}$ | 74 ± 8 |
| | | $EF_{CH_4}$ | 2.23 ± 0.23 |
| Vernooij et al. (2021) | Cerrado | $EF_{CO_2}$ | 1,664 |
| | | $EF_{CO}$ | 48 |
| | | $EF_{CH_4}$ | 0.78 |

### 3.2.3 Fire behavior and intensity

Fire behavior is limited by fuel characteristics and availability, and microclimate conditions. Fire behavior is often analyzed in terms of fire intensity (Gomes et al., 2020a; Silva et al., 2021), fire spread (Gomes et al., 2020a), heat released (Gomes et al., 2020a), fuel consumption (Andela et al., 2016), and fire return interval (Gomes et al., 2020b; Pereira Junior et al., 2014). Fire intensity here is defined as the rate at which fire releases heat energy (DeBano et al., 1998). For the Cerrado, this means that fire intensity follows a seasonal pattern, increasing in the dry months (Silva et al., 2021), and that it is also highly influenced by the vegetation type, increasing from forests to savannas and grasslands, where fine fuel consumption is higher (Gomes et al., 2020a). Silva et al. (2021) indicates higher values of fire intensity at the end of the dry season in the Cerrado, when fuel moisture is lower and fuel availability for burning is higher.

Fire intensity can be measured through the fire radiative power (FRP). FRP is the instantaneous amount of energy released by fire in the combustion process (Wooster, 2002). FRP often derives from MODIS data, and it relates to the intensity of fire and to the amount of biomass being consumed (Wooster, 2002). Although FRP has been used to provide estimates of fire intensity, Sperling et al. (2020) states that FRP from MODIS is underestimated for the Cerrado. Through FRP, Silva et al. (2021) estimates fire intensity in the Cerrado, with high values (FRP > 63.7 MW) found in the border with other biomes.

FRP data was combined by Andela et al. (2016) to derive fuel consumption estimates, which depends on combustion completeness and amount of fuel available for burning. For the savannas in South America, where the Cerrado is included, high values of fuel consumption were estimated compared to other savannas (Andela et al., 2016). For Andela et al. (2016), fuel consumption is partly driven by fire return periods, with grasses favoring short return interval, and thus resulting in low fuel build-up rates and lower fuel consumption. Pereira Junior et al. (2014) modelled fire return interval for a Protected Area in the Cerrado from 1997–2008, finding fire return intervals of three to six years, depending on the vegetation type. Shorter fire return periods (biennial) have been reported for other areas of the Cerrado (see Batista et al, 2018; Gomes et al., 2020b), typically related to the fine fuel load.

Fire spread in Cerrado is connected to fuel load and moisture (Gomes et al., 2020a; Silva et al., 2021). Fires in the Cerrado spread from savannas and grasslands to forests due to higher fine fuel load available for burning, and the same pattern is observed for heat released (Gomes et al., 2020a). These reflect combustion completeness and carbon emission rates of the different physiognomies in the biome (Gomes et al., 2020a).

Together, these findings emphasize the critical role that specific fire dynamics parameters play in shaping fire emissions within the Cerrado. To estimate carbon emissions from fire, researchers utilize various methodologies and data sources to quantify the carbon released during combustion. By understanding how burned area, fuel load, combustion completeness, combustion efficiency and emission factor vary across the landscape and seasons, researchers can better estimate fire emissions and their contribution to global budget. The variables identified in this review as predominant factors associated with fire emissions in the Cerrado are summarized in Fig. 6. In the next sections, we discuss how current research brings this information together to estimate fire emission in the Cerrado.

430

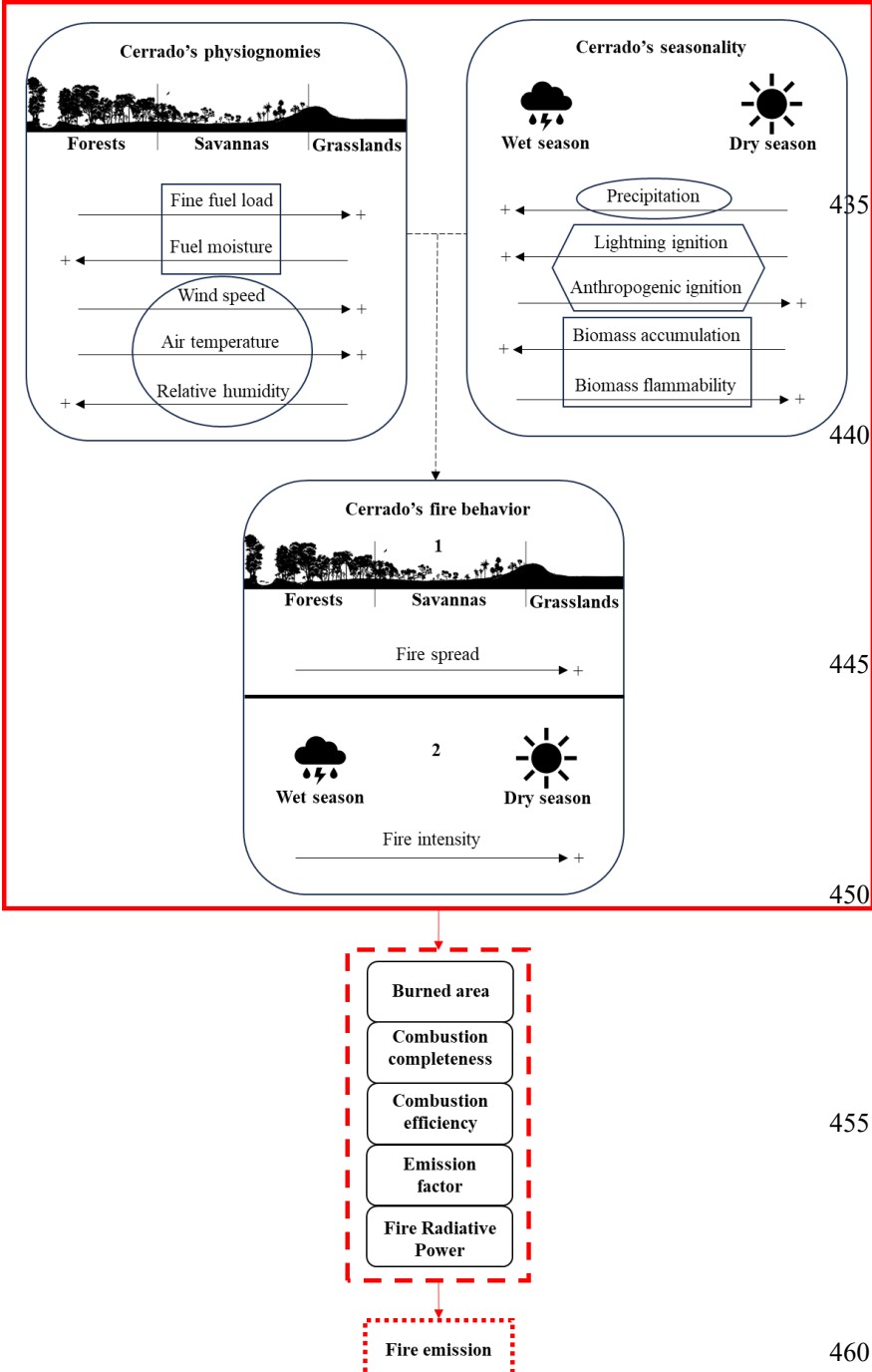

**Figure 6: Variables associated with estimating fire emissions in the Cerrado found in the literature. The Cerrados's physiognomies, separated into forests, savannas and grasslands, increase in fine fuel load and decrease in fuel moisture from forests to grasslands. Microclimatic conditions also change across the physiognomies, with increasing wind speed and air temperature, and decreasing**

relative humidity from forests to grasslands. The Cerrado's seasonality is divided into wet and dry seasons. The wet season is characterized by high precipitation, lightning ignitions and accumulated biomass, whereas the dry season is characterized by low precipitation, anthropogenic ignitions and flammable biomass. Fuel characteristics (square black boxes), climatic conditions (circle black boxes) and ignition (hexagon black boxes) interact (dashed black lines) to determine the Cerrado's fire behavior. Two aspects of fire behavior are presented (numbers 1 and 2): 1) fire spread increases from forests to grasslands; 2) fire intensity increases in the dry season. The Cerrado's physiognomies, seasonality and fire behavior together (red solid square) interact to determine the size of burned area, combustion completeness, combustion efficiency, emission factor and FRP. These (red dashed line) drive the resultant fire emissions (red dotted line). The image representing the Cerrado's physiognomies was adapted from the Brazilian Agricultural Research Corporation (Embrapa, 2024).

### 3.3 Estimated fire carbon emissions in the Cerrado

Papers under 'emission estimates' account for 43% (43 papers, due to papers double counted) of the papers reviewed, with 23 papers focusing on estimating emissions alone, while the remaining papers include a combination of emissions estimates, fire dynamics parameters and/or fire management and policy. Given the importance of carbon gases released into the atmosphere during combustion to the global carbon budget, mainly in the form of carbon dioxide ($CO_2$) (Ward and Hardy, 1991), this section will synthesize the main findings related to fire carbon emissions from the Cerrado found in the literature review process, notwithstanding the impact of non-carbon gases emitted during biomass burning.

The amount of carbon emitted to the atmosphere by fires is typically inferred by models implemented at different scales ranging from local to global analyses – of the 43 papers under 'emission estimates', 25 handle modelling techniques. Global and regional analyses tend to be less detailed, as they usually focus on capturing absolute emissions and on studying general aspects of large areas through a coarse resolution, and these are necessary to assess the impact of emissions on the global carbon balance (Palacios-Orueta et al., 2005; Rabin et al., 2018).

Two global emissions datasets were often used in the papers reviewed to develop and evaluate models of fire occurrence and effects, and these are GFED and Global Fire Assimilation System (GFAS). Both rely on MODIS products to estimate emissions. GFED fire emissions estimates are based on MODIS burned area products and on the Carnegie–Ames–Stanford Approach (CASA) model (Van Der Werf et al., 2017). GFAS estimates fire emissions globally by using a conversion factor that links MODIS-derived FRP observations to combustion rates, resulting in the fuel consumption, which is then combined with emissions factors to estimate fire emissions (Kaiser et al., 2012).

For example, Burton et al. (2021) uses the Interactive Fire and Emission Algorithm for Natural Environments (INFERNO) integrated into the Joint UK Land Environment Simulator (JULES) to evaluate current and future fire emissions in South America, and results are compared to both GFED and GFAS. Pereira et al. (2016) estimate fire emissions also in South America using the Brazilian Biomass Burning Emission Model with FRP assimilation (3BEM_FRP), and results are also compared to GFAS.

From 1997 to 2016, GFED fire emissions averaged 2.2 PgC year$^{-1}$ globally (Van Der Werf et al., 2017). In the SHSA region, where the Cerrado is located, fire emissions averaged 0.291 PgC year$^{-1}$, of which 49.3% were from savanna fires (0.14 PgC year$^{-1}$). Considering these values, savanna fires from SHSA, which broadly include the Cerrado, account for 6.36% of

annual global total carbon emissions from fires. GFAS does not provide subregional analysis, but carbon emissions for South America averaged 0.349 PgC year$^{-1}$ (2003–2008), while GFED version 3.1 averaged 0.299 PgC year$^{-1}$ (Kaiser et al., 2012).

This literature review identified 4 papers dedicated to estimating fire carbon emissions from the Cerrado fires, being one exclusively focused on estimating fire-related carbon emissions from the Cerrado (Gomes et al., 2024), while the other studies either include the Cerrado, but are not restricted to it (da Silva Junior et al., 2020), or include emissions as well as fire dynamics parameters (Gomes et al., 2020a; Oliveira et al., 2021). Yet, some studies that estimate fire emissions in the Cerrado are not focused on carbon emissions, but on aerosol (e.g. Hodgson et al., 2018; Pereira et al., 2009; 2022), fine particulate matter ($PM_{2.5}$) (e.g. Mataveli et al., 2019; 2023; Santos et al., 2021) and nitrogen compounds (e.g. Pope te al., 2020).

Gomes et al. (2024) uses satellite-derived burned area maps, surface fuel material maps, combustion factors, and emissions factors to estimate net $CO_2$ emissions from the Cerrado fires, among other greenhouse gases. From 1985 to 2020, the Cerrado savanna emitted approximately 2,227,964 Gg of $CO_2$ (0.018 PgC year$^{-1}$) and removed approximately 1,495,725 Gg of $CO_2$ (0.011 PgC year$^{-1}$) due to biomass regrowth, resulting in a net $CO_2$ emission of 859,701 Gg (0.007 PgC year$^{-1}$) (Gomes et al., 2024). From GFED4s and MODIS data, da Silva Junior et al. (2020) estimated carbon emissions for the Brazilian biomes between 1999 and 2018. Brazilian biomes produced 8.09 PgC of fire emissions (equivalent to 0.40 PgC year$^{-1}$). By analyzing all the Brazilian biomes, da Silva Junior et al. (2020) put fire emissions in the Cerrado into a national perspective, where it contributes 32.04% of total fire emissions (about 0.13 PgC year$^{-1}$), similar to the values found by Van Der Werf et al. (2017) for the savanna fire emissions in the SHSA region. According to da Silva Junior et al. (2020), the Cerrado is a major contributor to Brazil's fire emissions.

Gomes et al. (2020a) modelled carbon emissions associated with fine fuel consumption, finding 0.230 kg m$^{-2}$ for grassland, 0.210 kg m$^{-2}$ for savanna, and 0.053 kg m$^{-2}$ for forests, and concluding that fine fuel load was the main predictor of the amount of carbon released through fire. When considering different scenarios (moderate, medium, and extreme) for fine fuel available for burning, wind speed, and vapor pressure deficit using the BEFIRE (Behavior and Effect of Fire) model, Gomes et al. (2020b) showed that carbon emissions from fine fuel consumption increased with the intensity of the scenario (0.19 kg m$^{-2}$ for moderate, 0.23 kg m$^{-2}$ for medium, and 0.26 kg m$^{-2}$ for extreme). Because the model only considers fine fuel load, which is more abundant in grasslands due to the presence of grasses, carbon emissions decrease with the increase of woody biomass. These simulations confirm that fire-associated emissions depend on the vegetation type (Gomes et al., 2020a, 2020b).

Oliveira et al. (2021) modelled fire emissions across the Cerrado by estimating fuel loads through remote sensing data over four years (2015–2018). Results averaged 0.066 ± 0.013 Pg $CO_2$ year$^{-1}$ (0.018 ± 0.00354 PgC year$^{-1}$). When accounting for regrowth uptake, net emission was 0.015 ± 0.004 Pg $CO_2$ year$^{-1}$ (0.00487 ± 9.65 $10^{-4}$ PgC year$^{-1}$). Oliveira et al. (2021) consider these values low and suggest incorporating a more detailed vegetation map and the burning intensity of different fuel types to improve their estimates. The values of fire carbon emissions restricted to the Cerrado reported in this section are summarized in Table 2.

**Table 2: Values found in the literature for fire carbon emissions in the Cerrado. Units are: petagrams of carbon per year (PgC year[1]) and kilogram per square meter (kg m[-2]).**

| Study | Study area | Value | Units | Observation |
|---|---|---|---|---|
| da Silva Junior et al. (2020) | Cerrado | 0.13 | Pg C year$^{-1}$ | Based on MODIS data and on GFED4s. |
| Gomes et al. (2020a) | Cerrado – Grassland | 0.23 | Kg m$^{-2}$ | The study estimates the amount of carbon released in combustion, used as a proxy for estimates of fire-associated emissions. |
| | Cerrrado – Savanna | 0.21 | Kg m$^{-2}$ | |
| | Cerrado – Forest | 0.053 | Kg m$^{-2}$ | |
| Oliveira et al. (2021) | Cerrado | 0.018 ± 0.00354 | Pg C year$^{-1}$ | Includes results from a bayesian model developed by the authors based on remote sensing imagery, as in Franke et al. (2018). |
| Gomes et al. (2024) | Cerrado – Savanna | 0.018 | Pg C year$^{-1}$ | Focuses only on the savanna formation of the Cerrado. |

The difference in values (Table 2) indicates the complexity of estimating fire emissions, especially in a diverse region such as the Cerrado. The estimation of fire emissions relies on multiple variables, each quantified by different methodologies and databases. Also, the procedure used to extrapolate measurements and estimate emissions on broader scales differs among studies. When combined, these variabilities result in uncertainties associated with estimating fire emissions, often reported by the studies synthesized in this review (e.g. Andreae, 2019; Oliveira et al., 2021; Vernooij et al., 2023).

### 3.4 The influence of fire management and policy in estimating fire emissions in the Cerrado

In synthesizing the literature on fire emission in the Cerrado, we identified 12% (12 papers) of papers focused on fire management and policy, most (8) under the 'review' and 'perspective' categories. This indicates that fire management and policy are important in understanding fire dynamics in the Cerrado. Still, papers that address these do not usually bring new information based on observation or experiments but tend to synthesize or opine on existing literature. For example, this review captured only one study that quantifies the amount of fire emissions mitigated by fire management in the Cerrado (Franke et al. (2024), probably due to the difficulty in quantifying the social and cultural aspects of fire, which are intrinsic to fire management and policy.

Fire exclusion policies arose in Brazil as a reaction to the misuse of fire for pasture management and deforestation, especially in the 20th century (Durigan and Ratter, 2016). These policies dominated for decades in the Cerrado (Durigan and Ratter, 2016), also due to the misbelief that fire harms the biome, resulting in fuel accumulation and changes in fire season and regime (Pivello, 2011; Pivello et al., 2021). Consequently, extensive LDS wildfires replaced small patchy burns (Moura et al., 2019; Pivello, 2011). A shift towards recognizing fire as essential to maintaining the Cerrado's diversity and ecosystem

services led to a change in the Federal Legislation in 2012 (Federal Law 12,651/2012) to explicitly allow the use of fire for ecological purposes in fire-prone settings (Durigan, 2020; Pivello et al., 2021). Further, the acknowledgment of controlled fires an essential socio-ecological component in fire-dependent ecosystems led to the Brazilian Integrated Fire Management
(IFM) Policy (Pivello et al., 2021), sanctioned in 2024 (Federal Law 14,944/2024).

The changes in policy to recognize the importance of fire to the Cerrado is also reflected in the increased numbers of Protected Areas (PAs) under fire management – the Pilot IFM project, named the *Cerrado-Jalapão*, started in 2014 in three Protected Areas. The *Cerrado-Jalapão* project improved the understanding of fire dynamics in the biome (Durigan and Ratter, 2016). In 2024, Franke et al. (2024) identified 36 PAs undergoing IFM activities. Similarly, Santos et al. (2021) documented
an increase in the number and spatial distribution of prescribed burning in PAs within the Cerrado from 2015–2018. Prescribed burning is a common activity in fire management used to reduce fuel load in fire-prone settings around the world (Santos et al., 2021).

Despite the increasing recognition of fire's importance to the Cerrado and the subsequent expansion of fire management operations, as well as the global relevance of fire management in reducing emissions (Moura et al., 2019), this
review identified only one study that quantifies the reduction in fire carbon emissions achieved through fire management in the Cerrado. Franke et al. (2024) show an emission abatement of 26,677 $tCO_2e$ year$^{-1}$ (2014 – 2019) in specific protected areas, and a reduction potential of more than 1,085 $tCO_2e$ year$^{-1}$ (2014 – 2019) when the result is scaled-up to all protected areas in the Cerrado. The reduction in emissions from prescribed burning is due to lower fuel consumption and combustion factors of early dry season fires when compared to mid/late dry season fires (Franke et al., 2024). These values are considered
conservative due to analyzing mainly fine fuels, and Franke et al. (2024) argue that estimations could be improved by using high-resolution data that would allow the identification of small-scale fires.

This literature review also documents studies that estimate activities associated with fire management, such as prescribed burning, important for further emission estimates. Mistry et al. (2019) suggests reduced burned area and late dry season emissions from IFM in the Cerrado. Batista et al. (2018) compares two areas within the Canastra National Park, one
under fire suppression and the other under fire management. The area under fire management presented less burned area annually (2000–2015), with a higher proportion of areas burned in the EDS. Similarly, Franke et al. (2024) identified increased EDS and decreased LDS burned area extents as a result of fire management. Over 2014–2019, Franke et al. (2024) estimated an average of 20.4% EDS burned area in PAs under IFM, against 4.7% in PAs that do not apply IFM. Conversely, middle to late dry season burned area averaged 76.5% in IFM PAs, while that average reaches 87.1% in non-IFM PAs. Santos et al.
(2021) also documents reduced burned area in the late dry season due to fire management in two Indigenous Territories in the Cerrado, which led to reduced fire intensity and reduced extreme wildfires, indicating a reduction in further fire emissions.

The literature reviewed shows that assessing fire emissions in the Cerrado is particularly complex due to the region's ecological diversity and the interplay of fire dynamics, policy, and cultural practices. Fire management remains challenged by the lack of understanding of fire dynamics parameters from early and late dry season burning (Mistry et al., 2019; Pivello et
al. 2021), as estimated biomass, combustion completeness and burned area, and these affect fire emissions (Mistry et al., 2019).

Understanding the role of these aspects in estimating fire emissions contributes to the development of consistent fire policies in Brazil (Durigan, 2020; Durigan and Ratter, 2016; Moura et al., 2019), which influence Brazil's national and international commitments to carbon emission reductions (Pivello et al., 2021).

**4 Discussion**

To our knowledge, and according to our search criteria, this is the first systematic literature review to provide an overview of fire emissions in the Cerrado. By analyzing existing literature on fire emissions in the Cerrado, we identified key topics that contribute to a broad and holistic understanding of the role of these emissions in the carbon budget on regional, national and global scales. This understanding includes not only direct fire-related carbon emission, but also the underlying 610 fire dynamic parameters, fire management practices, and fire policies, along with the various methodological approaches used to estimate these.

In synthesizing literature, we observed a growing interest in fire science in the Cerrado from multiple perspectives, as shown in the number of papers published annually. Pivello et al. (2021) also observes an increase of studies on fire. This is probably due to an increased recognition of fire's importance in the global carbon balance and the increased number of alarming 615 fire events reported in recent years due to climate change (Burton et al., 2024; Hofmann et al. 2021; Oliveira et al., 2022). The year 2022 did not follow the growth trend, which could indicate a gap in publications this year or a limitation of our research method that could not capture publications in 2022. The following years tend to increase publications compared to 2022. The decrease in 2022 could also indicate a shifted focus away from the Cerrado studies due to political or financial constraints to encourage scientific studies in the region, or due to a shifted focus towards other regions of Brazil. Pereira et al. (2024) indicate 620 an increase in papers published about fires in the Pantanal after the 2020 megafire in the biome. Papers related to fire dynamics and emissions in the Pantanal published in 2022 show fire as a consequence of the compound impact of land use and climate in these regions, and similar findings were reported for the Amazon rainforest in papers also published in 2022 (see Barbosa et al., 2022; Dutra et al., 2022; Menezes et al., 2022; Silva et al, 2022; Walker et al., 2022).

We identified that papers often encompass a global and regional emissions analysis, with local (Cerrado) analysis 625 accounting for about 36% of papers included in this review. The remaining 64% of papers represent global and regional (Brazil, South America, Tropical region) analysis, often understanding the role of emissions from a holistic perspective and providing insights into the influence of local emissions on the global carbon balance. Additionally, many papers in this systematic literature review are led by non-Brazilian institutions or lack authors from Brazilian institutions. In fact, fire emissions studies in the Cerrado are rarely led by institutions within the region. This highlights opportunities to enhance collaboration between 630 Brazilian and non-Brazilian institutions, and stronger partnership between different regions within Brazil.

The papers reviewed have shown that input data uncertainty affects output accuracy. In the Cerrado's studies, for example, uncertainties regarding the accuracy of spatial patterns of physiognomies and climatic seasonality throughout the biome (i.e., length of dry and wet seasons and rainfall amount) impact the absolute estimates of fire emissions and future projections. While global-scale studies often generalize such patterns, local studies include the complexity and diversity of a

limited region, which is essential to capture changes in fire dynamics and to assess the components that influence emissions at smaller scales (Palacios-Orueta et al., 2005). They might be extrapolated to represent more extensive areas. Although there are uncertainties with both global and local scales, the use of remote-sensing techniques contributes to the accuracy of emission estimations, and it is the core of much recent research regarding fire occurrence and emissions worldwide (Lasslop et al., 2019). However, it is important to acknowledge that some uncertainties, such as limitations associated with spatial resolution

and with expanding the details of local results to a broader scale, may also limit the accuracy of results, as observed in fire management potential assessments (Griscom et al., 2020).

Some changes that can be made to improve carbon accounting from fire in the Cerrado include: acknowledging the heterogeneity of the biome, especially regarding climatic seasonality and fuel characteristics (Gomes et al., 2018; Oliveira et al., 2021); incorporating location-specific algorithms and datasets that improves the representation of the Cerrado (Libonati et

al., 2015; Mataveli et al., 2024; Oliveira et al., 2021); and accounting for other carbon pools, such as soils and belowground, which are large components of carbon in the Cerrado's physiognomies (Bustamante et al., 2012).

Examining fire carbon emissions reveals that local emissions reflect the global carbon budget. A key factor in carbon balance analysis is vegetation regrowth, since a significant portion of the $CO_2$ emitted by fire is sequestered during post-fire biomass recovery (Andreae, 2019; Van Der Werf et al., 2017; Gomes et al., 2024). This literature review identified one study

that includes the removal of $CO_2$ by regrowth in the Cerrado, quantifying the net $CO_2$ emissions from the Cerrado fires from 1985–2020 (Gomes et al., 2024). Vegetation regrowth removed 63.5% of the $CO_2$ emitted, indicating that fire in the Cerrado has been a source of carbon to the atmosphere in recent decades (Gomes et al., 2024). For a shorter time series (2015–2018), Oliveira et al. (2021) also found the Cerrado fires to be a net emitter of $CO_2$. Further research is needed to enhance the understanding of the long-term carbon balance of Cerrado fires. This literature review contributes by providing an overview

of published studies on fire emissions in the region.

Compiling literature on fire emissions in the Cerrado has revealed several studies that focus on fire dynamics, management, and policy rather than estimating emissions, with studies often stating their relevance in providing insights into emission estimates and the importance of the Cerrado's fire emission in the global carbon balance. This indicates that estimating fire emissions requires a holistic approach. For example, including the perspectives of fire culture, ecology and

policy within emissions is essential given the importance of fire to the biome. Fire culture refers to the interaction between humans and fire, encompassing the factors that drive societies to use it (see Pivello et al., 2021). The use of fire, shaped by cultural traditions and socioeconomic conditions, can influence the extent of burned areas and the resulting fire emissions. Traditional communities, for instance, may occasionally use fire on a small scale (Pivello et al., 2021).

Coupled with the available biomass for burning and its characteristics — which depend on vegetation type, density,

moisture and seasonal growth patterns — burned area, fuel characteristics, combustion completeness, combustion efficiency and emission factor set the stage for potential emissions. Fire intensity, driven by conditions such as dry weather, strong winds, and fuel accumulation, influences combustion efficiency. High-intensity fires tend to consume more fuel, resulting in higher combustion efficiency and more complete combustion. This reduces emissions of pollutants such as carbon monoxide and

particulate matter but increases emissions of carbon dioxide. In contrast, incomplete combustion results in higher emissions of pollutants such as particulate matter and carbon monoxide, which may persist in soils over long periods. Combustion completeness further influences the amount of biomass converted to carbon and released into the atmosphere. Together, these parameters allow for the estimation of emissions based on the combination of burned area, fuel load, and combustion completeness.

Thus, this review indicates a critical need to develop interdisciplinary studies to bridge fire policies and fire emissions in the Cerrado. Understanding fire dynamics, including the opportunities for mitigating emissions from fire activities, is essential for recognizing fire's role in achieving global environmental and climate targets. For instance, Martin (2019) identifies United Nations Sustainable Development Goals that are related to fire and land management, as goals 3 (good health and well-being), 13 (climate action), and 15 (life on land). These impact the 2015 Paris Agreement target to limit warming to 1.5 °C by 2100. In fact, da Silva Junior et al. (2020) find that fire activities undermine Brazil's emission reduction targets under the Paris Agreement. The Paris Agreement outlines commitments for climate actions and acknowledges the importance of mitigation and removal actions, where fire management can play an important role. If great effort is put into mitigating emissions and removing carbon, the 1.5 °C could be achieved, although ambitious, with Brazil holding the highest mitigation potential in the land sector (Roe et al., 2019). Together with other countries, improved forest management in Brazil, which includes fire management, could be able to increase carbon removal by 40 $GtCO_2$ by 2050 (Roe et al., 2019).

Studies that discuss the cultural and socioeconomic aspects of fire often do not discuss them from the emissions' perspective. Estimating the influence of humans on fire emissions is a complex task, which is also reflected in the lack of equations and algorithms to reproduce fire management strategies in land surface models. This complexity emphasizes all factors that need to be considered beyond quantifying the amount of carbon emitted to the atmosphere. Despite the difficulty in measuring these, the lack of inclusion of these aspects in fire emission estimates could also be due to the shift towards recognizing fire as essential to the Cerrado being recent, especially when compared to other fire-prone settings. For example, the WALFA (West Arnhem Land Fire Abatement) project in northern Australia became entirely active in 2005 (Russell-Smith et al., 2013), where traditional people, scientists and governmental institutions collaborate to reduce fire emissions through fire management activities (Russell-Smith et al., 2013). Meanwhile, the Pilot IFM project in the Cerrado started in 2014 (Schmidt et al., 2018).

Compared with the Cerrado, a relatively high number of fire studies were performed in Australia, which demonstrate the potential to expand the study of emissions from fire in the Cerrado. Da Veiga and Nikolakis (2022) counted 64 papers from Australia and 29 from Brazil when documenting the interaction between fire management and carbon programs worldwide. The Australian studies, especially in the WALFA region, act as an example to other savanna environments, including to the Cerrado's Pilot IFM project (Schmidt et al., 2018). The WALFA project reaffirms the potential of management activities to reduce emissions in savanna countries. EDS burns in the region emit 48% of what is emitted in the LDS (Russell-Smith et al., 2009). Similarly, Khatun, Corbera, and Ball (2017) suggest that, in the Tanzanian miombo, EDS burns could avoid carbon

emissions and enhance carbon uptake. Studies in Mozambique and Botswana explore the potential of EDS burns to reduce emissions in southern African savannas (Russell-Smith et al., 2021).

Climate change increasingly affects fires, and adaptation and mitigation activities are essential to limit these effects (Burton et al., 2024). Direct human impacts may offset the effects of climate change in fire worldwide (Burton et al., 2024), especially in fire-prone environments. In fact, Andela et al. (2017) found a decreasing trend in fire activity driven by human activities worldwide. These indicate an opportunity to investigate the potential of fire management to mitigate emissions in the Cerrado, and to understand fire emissions in the biome. The lack of data availability and accessibility has been acknowledged in literature as a gap in research regarding fire dynamics in the Cerrado, primarily due to its heterogeneity

(Bustamante et al., 2018; Gomes et al., 2018; Oliveira et al., 2021). Developing more specific algorithms able to capture the climatic seasonality and fuel diversity of the Cerrado, understanding the above and belowground carbon pools of each physiognomy by in situ evaluations and satellite-derived approaches, and incorporating these into fire emissions estimates in the Cerrado could improve carbon measurements in the biome.

    Consequently, this would improve fire emission estimates in the Cerrado. Pathways towards this improvement include

connecting observational information with modeling and a better assessment and quantification of the impact of qualitative aspects in fire estimates. Examples of how these can be achieved are by better accounting of emission parameters across the Cerrado's vegetation types and fire regimes, valuing prescribed burning emissions and including these in fire modeling, representing fire management in land surface models, using in situ observations to assess models' utility and as input data to modeling, and incorporating the ecological, social and cultural aspects of fire in fire emission estimates. These could address

uncertainty and improve models' accuracy, thus providing better accounting of fire emissions in the Cerrado and worldwide.

    Our research question is "How compiling published material on fire emissions in the Cerrado can provide a holistic understanding of their role in the global carbon budget?". Analyzing published papers on fire emissions in in the Cerrado provides valuable insights into its role in the carbon balance. These include understanding the parameters used to estimate emissions, quantifying the amount of carbon released into the atmosphere by fires, and identifying important aspects of fire

dynamics that are sources of uncertainty or are not considered in fire emission estimates. These are summarized in Table 3.

**Table 3. Parameters included in current studies and parameters to be considered for future studies to estimate fire emissions in the Cerrado captured by this review.**

| Parameters included in current studies in the Cerrado | Parameters to be considered for future fire emission estimates in the Cerrado | |
|---|---|---|
| Biomass burned | Belowground and soil carbon pools | 730 |
| Burned area | Fire culture | |
| Combustion efficiency | Fire ecology | |
| Combustion factor | Fire policy | |
| Emission factor | Location-specific algorithms | |
| Fire intensity | Socioeconomic aspects of fire | |
| Fuel characteristics | | 735 |

This systematic literature review presents an overall assessment of published literature on fire emissions in the Cerrado to understand its placement in the carbon budget, considering the criteria used to narrow our search. Including Portuguese as a research language and consulting other search database platforms, such as Web of Science, could have resulted in more papers included. The "grey literature" was not in the scope of our research method but could potentially result in more

findings. "Grey literature" is defined as "*unpublished research (e.g., dissertations, conference abstracts, preprints or unpublished datasets), or those published outside of traditional academic publishing (e.g., governmental reports)*" (Foo et al., 2021, 1711).

Finally, the challenges in estimating fire emissions worldwide lie not only in measuring carbon directly released, but also in capturing the nuanced effects of fire on ecosystems and the broader Earth system (Hamilton et al., 2024). Continued

and diversified research is needed to improve the understanding of fire dynamics in the Cerrado and how these reflect fire emissions locally and globally. These will assess current knowledge gaps regarding fire emission estimates in the Cerrado by enhancing the understanding of the role of emissions from the biome in the global carbon budget and potential mitigation activities in achieving global environmental and climatic goals, and by providing a better grounding for future projections.

## 5 Conclusion

This systematic literature review synthesized 77 peer-reviewed papers, from local to global scales, according to a set of criteria to understand fire emissions in the Cerrado and their placement in the global carbon budget, from a holistic perspective. From our literature review process, we found that research on fire emissions in the Cerrado is still overlooked when compared to other savanna ecosystems. Based on our knowledge and search criteria, this is the first systematic literature review to provide an integrated understanding of fire emissions in the Cerrado, where fire dynamics, management and policy

emerge as crucial for estimating fire emissions. By providing necessary context on the drivers of emissions, their variability, and potential mitigation strategies, we highlight how insights from fire behavior and fire regimes are essential for estimating emissions.

Although our research question and keyword search focus on fire emissions, the literature review process often revealed papers on fire dynamic parameters that are drivers to emissions estimates, indicating a gap in connecting fire dynamics

and emissions research in the Cerrado. Thus, we examine how fire dynamics shape emissions and identify where further integration of these fields could improve our holistic understanding of fire emissions in the Cerrado. This is possible due to a clearer understanding of the variables considered when estimating emissions, a deeper comprehension of the published values estimated, and the identification of important aspects of fire regimes that influence fire policy and mitigation strategies.

Fire management and policy emerged as key themes in our literature search because they play a crucial role in shaping

fire regimes and, consequently, fire emissions in the Cerrado. While there is limited empirical research quantifying the direct impact of fire management on emissions in this region, discussing management strategies is essential to fully assessing the

factors that influence fire-driven carbon fluxes. Understanding how management affects fire regimes provides a pathway to improving emissions estimates and identifying potential mitigation strategies.

Thus, this review demonstrates that understanding the placement of fire emissions in the global carbon budget requires a holistic approach that draws together disciplines across fire science, especially in a distinct environment such as the Cerrado, while reinforcing the urgent need for further investigation into the topic. The complexity of estimating fire emissions outreach measuring carbon, and this review highlights the urgent need for interdisciplinary studies to connect fire parameters with fire emissions, which then influence fire policies and the achievement of climatic commitments. Continued research is needed to fully understand and quantify the influence of fire dynamics and mitigation strategies on fire emissions in the Cerrado.

**Code availability:** No codes were used in this literature review.

**Data availability:** The list of papers reviewed is available in the Supplement.

**Authors' contribution:** RMV conceptualized the study and prepared the original draft. RMV, CvR, CB, DIK, MC and FM interpreted and analyzed data. RMV, CvR, CB, DIK and MC substantially revised and edited the study.

**Competing interests:** The authors declare that they have no conflict of interest.

**Acknowledgments:** RMV thanks the São Paulo Research Foundation (FAPESP) for grants 2020/06470-2 and 2022/13322-5. CvR acknowledges FAPESP grant 2017/22269-2 and CNPq grant 314780/2020-3. CB was funded by the Met Office Climate Science for Service Partnership (CSSP) Brazil project which is supported by the Department for Science, Innovation & Technology (DSIT). DIK was supported by the Natural Environment Research Council as part of the NC-International programme [NE/X006247/1] delivering National Capability. MC acknowledges the support from the São Paulo Research Foundation (FAPESP, Process 2015/50122-0).

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
