# Peer review of "Review article: Fire emissions in the Brazilian Cerrado: dynamics, estimates, management and their role in the global carbon budget"

_EGUsphere, 2024_

## Author Comment (AC1)

**Response to comments by referee #1 on the manuscript egusphere-2024-2348**

We, the authors, thank the editor for handling the paper and the reviewer for their comments and suggestions. We value the careful feedback provided, and we believe this is important for improving the quality of our review paper. We provide a table with detailed responses to each separate comment. According to the editor's instructions, the revised manuscript should not be prepared at this stage. Therefore, we have not included the specific line(s) or page(s) where changes were made, nor the updated figures and tables. Instead, we have provided the edited or added paragraphs and sentences to demonstrate how we have addressed the reviewer's comments.

Sincerely,

Renata Moura da Veiga (on behalf of all co-authors)

| Reviewer 1 | |
|---|---|
| **Comment** | **Authors' response** |
| The question that the authors use to guide their review process is the following: "How compiling published material on fire emission in natural areas of the Cerrado can provide a better understanding of the placement of these emissions in the atmospheric carbon budget?". This question is not mentioned again in the manuscript and it is left unanswered in the Discussion. Additionally, its formulation is not in line with the main goals of the manuscript. | We have now mentioned the question again, and have answered the question more directly, in the discussion section. The following new paragraphs are meant to complement the information that was already in the manuscript:

"Our research question is "How compiling published material on fire emissions in areas of the Cerrado that do not explicitly include anthropogenic land uses can provide a better understanding of the placement of these emissions in the global carbon budget?". Analyzing published papers on fire emissions in these areas in the Cerrado provides valuable insights into its role in the carbon balance. This includes understanding the parameters used to estimate emissions, quantifying the amount of carbon, especially $CO_2$, released into the atmosphere by fires, and identifying important aspects of fire dynamics that are sources of uncertainty or are not considered in fire emission estimates. These are summarized in Table 3. |

| | Aiming at compiling literature on fire emissions in the Cerrado has led to a number of papers that do not explicitly estimate fire emissions itself, but rather discuss fire dynamics and parameters used to estimate emissions. This indicates that there is a gap in the literature regarding fire emissions estimates in the Cerrado. However, studies have indicated that fires in the Cerrado play an important role in the global carbon balance. For example, Van Der Werf et al. (2017) found that savanna fire emissions from the Southern Hemisphere South America region, which includes the Cerrado, averaged 0.14 PgC year$^{-1}$ over 20 years, accounting for more than 6% of global fire emissions per year. Similarly, and from a national perspective, da Silva Junior et al. (2020) have shown Cerrado fires contribute more than 32% of the Brazilian total fire emissions (about 0.13 PgC year$^{-1}$ over the 20 years)." |
|---|---|
| For instance, the question refers to "natural areas of the Cerrado". If this were to mean areas of intact native vegetation, the authors would need to provide a keyword for this, as the vast majority of papers that are mentioned do not focus on natural areas, and are often estimates for the entire biome or specific land cover types. | Regarding the "natural areas" limitation - this means that we have excluded papers that explicitly include anthropogenic land uses. We have made this clearer in the research question and when explaining the inclusionary criteria: "After establishing our research question as "How compiling published material on fire emissions in areas of the Cerrado that do not explicitly include anthropogenic land uses can provide a better understanding of the placement of these emissions in the global carbon budget?" "We applied four inclusionary criteria to identify relevant literature: papers had to be (1) published in peer-reviewed journals with an impact factor greater than 1; (2) encompass the Cerrado biome; (3) be published after 2003, for a two-decade period (2003-2022); and (4) be conducted in areas that do not explicitly include anthropogenic land uses, here referred to as "natural areas". We define natural areas as those covered by natural vegetation of the |

| | Cerrado, where anthropogenic land uses do not occur (e.g. pasture and agriculture). According to this criteria, 48.66% (965,783 km2) of the Cerrado is covered by natural vegetation (MapBiomas, 2022). Because papers found by this review often do not specify the land use of their study area, we have not included papers that explicitly document fire occurring in anthropized areas, as a proxy for documenting existing literature on natural areas of the Cerrado. To improve the assessment of our research question, we have also incorporated in our review papers that don't focus only on the Cerrado, but rather include it as part of the analysis." |
|---|---|
| I also believe that "global carbon budget" would be more appropriate than "atmospheric carbon budget". | We have replaced "atmospheric carbon budget" for "global carbon budget", including in the title. |
| This is a major concern, as the papers found through the PRISMA method are never listed. The authors refer to many papers throughout the text, but the reader does not know if these papers are those included in the literature review, or just part of a discussion. There is no list, even in Supplementary Material, of the papers, along with their respective topic (fire dynamics parameters, emission estimates, and fire management and policy) and study design (empirical, review, and perspective). | The 69 papers reviewed will be included as a table in the supplementary material. The columns included are: paper title, year of publication, authors, area of study, topic, methodological technique, study design. |
| Moreover, these classifications are explained in the Results section (e.g. lines 210-214) rather than in Methods. The authors also divide the papers according to study area (Global, Tropical region, South America, Brazil, Cerrado) which is never mentioned in Methods. | We have outlined what the study areas are in the Methods section: "We classified the reviewed papers based on (a) location range, from global to local scales: global, tropical region, South America, Brazil and Cerrado". |
| There are many papers, especially in the "fire dynamics parameters" category, that do not evaluate emissions. Although burned area and fire intensity are parameters used | The aim of our review is to provide a comprehensive view of the knowledge and gaps related to fire emissions in the Cerrado, specifically focusing on impacts on the carbon |

| to estimate emissions, discussing fire patterns and their climatic and human drivers should not be a main focus of this literature review. | cycle. |
|---|---|
| | To clarify, while we emphasize emissions, we view fire dynamics (such as burned area and fire intensity) as essential parameters that support our understanding of emissions. They represent important "tools" in our framework, helping us interpret the conditions under which emissions are generated and their variability due to climatic, ecological, and human influences. Additionally, fire emissions data are often limited, and the literature reflects more studies on patterns and drivers, which provides useful background to identify gaps. |
| | • The influence of fire parameters in estimating fire emissions is also stated in the following paragraph (specifically in the bold sentence) in the Results section: |
| | "Of the 69 papers reviewed, 37 relate to fire dynamics parameters used to estimate emissions, 40 report the amounts of fire emissions, and 7 report fire management and policy - the total does not round up to 69 because 15 papers are related to more than one topic. **These numbers indicate that many papers are not related to reporting emissions but provide information to support the understanding and estimation of fire emissions, demonstrating a potential to expand the study of GHG emissions from fire in the Cerrado**. For example, Santos et al. (2021) use satellite imagery to estimate emissions and parameters, such as burned area and fire intensity, to support the application of prescribed burning in the Cerrado, but actual emissions estimates are not included." |
| | • Further, we have added the following sentences in the discussion and in the conclusion section to emphasize this point: |
| | "Aiming at compiling literature on fire |

emissions in the Cerrado has led to a number of papers that do not explicitly estimate fire emissions itself, but rather discuss fire dynamics and parameters used to estimate emissions. This indicates that there is a gap in the literature regarding fire emissions estimates in the Cerrado."

"This review demonstrates that papers fail to report on fire emissions itself, with fire dynamics and parameters used to estimate emissions in the Cerrado often being the focus of published literature."

- Additionally, we have done major revisions to the "Fire dynamics parameters to estimate fire emissions" section. It now contains introductory paragraphs to make this point clear, and it is further divided into subsections: Burned area and fuel characteristics; Combustion efficiency, combustion completeness and emission factor; Fire behavior and intensity.

The introductory paragraphs are:

"Papers on 'fire dynamic parameters' account for 44% of the studies reviewed, underscoring the importance of variables like burned area, fuel characteristics, combustion completeness, combustion efficiency and emission factor in fire emissions research. These parameters directly influence emission estimates, with their combination playing key roles in determining carbon emissions from fires. By examining these variables within the specific ecological and climatic context of the Cerrado, we gain insights into how fire behavior and emissions in this biome interact.

The area burned, typically measured via satellite or ground surveys, is one of the primary parameters for estimating emissions (Libonati et al., 2015; Mangeon et al., 2016; Silva et al. 2021). Coupled with the available biomass for burning and its characteristics —

which depend on vegetation type, density, moisture and seasonal growth patterns — these elements set the stage for potential emissions. Fire intensity, driven by conditions such as dry weather, strong winds, and fuel accumulation, influences combustion efficiency. High-intensity fires tend to consume more fuel, resulting in higher combustion efficiency and more complete combustion. This reduces emissions of pollutants such as carbon monoxide and particulate matter but increases emissions of carbon dioxide. Combustion completeness further influences the amount of biomass converted to carbon and released into the atmosphere. In contrast, incomplete combustion results in higher emissions of pollutants such as particulate matter and carbon monoxide and produces pyrogenic carbon, which may persist in soils over long periods. Together, these parameters allow for the estimation of emissions based on the combination of burned area, fuel load, and combustion completeness.

The prevalence of studies on these fire dynamics parameters reflects both the accessibility of these variables and a gap in linking fire dynamics directly to emission. This focus on fire dynamics provides some of the most current information available, yet it suggests a need for more research to fill the gaps in understanding the chain from fire drivers to emissions. We further discuss the fire dynamics parameters found in the literature review process."

- We have also updated aims a and b of the paper to make this clearer: "Thus, this systematic literature review synthesizes published material on fire emissions in areas of the Cerrado that do not explicitly include anthropogenic land uses, with aims to: (a) outline current emissions estimates, specifically $CO_2$, or fire dynamic factors that help support these estimates, in regions that

| | |
|---|---|
| | encompass Cerrado or are limited to it; (b) understand how these estimates fit the carbon budget". |
| The authors also select burned area as the sole parameter to estimate fire emissions in Figure 5 (Section 3.1), to then explain how Fire Radiative Power (FRP) can also be used in the last paragraph of Section 3.2. This shows a lack of grasp of some of these concepts: for instance, the authors introduce FRP in Section 3.2 as if it was not the same parameter as "fire intensity" in Section 3.1; they also mention that "FRP considers (...) area affected by fire" and that it uses "MODIS active fires are inputs" which is, at best, vey badly worded. | • FRP was moved to the "Fire behavior and intensity" subsection.

• Figure 5 was adjusted to include a combustion completeness box beside the burned area box, coming from fire behavior and seasonality. Both burned area and combustion completeness result in fire emissions. This figure is also in the "Fire behavior and intensity" subsection. |
| The Introduction fails to provide background to the importance of fire emissions in the Cerrado, both in the national and global context. The role of fire emissions in the global carbon cycle/budget should also be highlighted, along with the role of Brazil in the LULC emissions as the highest emitter (see the Global Carbon Budget 2023).

Information on how carbon emissions are estimated worldwide should be included (e.g. what data and methods are usually employed), so that the reader can better understand results found in the literature review. Moreover, as a tropical savanna, the Introduction could also leverage on information from other tropical savannas worldwide. | Thank you for your comment. We have already provided extensive background on Cerrado's role in emissions nationally and globally in the Introduction, especially in the following paragraph:

"From 1997-2016, savanna fires from Southern Hemisphere South America, which the Cerrado dominates, accounted for 6.36% of the global carbon from fires annually (Van Der Werf et al., 2017). This contribution is substantial, as it highlights the Cerrado's role as one of the world's major fire-emitting ecosystems. To put this into perspective, savanna fires from the Australia and New Zealand region, which refer to the Australian savanna, account for 4.55% of the global carbon from fire emissions emitted each year for the same period (Van Der Werf et al., 2017). Compared to the Cerrado, a relatively high number of fire studies are performed in Australia. Da Veiga and Nikolakis (2022) counted 64 papers from Australia and 29 from Brazil when documenting the interaction between fire management and carbon programs worldwide."

However, we have added more detail in |

| | response to this point, including the number for LULC emission for comparison: |
|---|---|
| | "The Cerrado's fires are potentially responsible for more than 30% (about 0.13 PgC year$^{-1}$) of Brazil's total fire emissions (da Silva Junior et al., 2020). As a comparison, the Cerrado accounts for about 14% of Brazil's emission from land use and land cover change (SEEG, 2023). Brazil is the highest emitter in the world in this category (Friedlingstein et al., 2023), contributing with up to 0.4 PgC year$^{-1}$ (Rosan et al., 2021). The Cerrado's role in Brazil's overall emissions profile is, therefore, critical, with fires contributing to a substantial share of the country's fire emissions, which has national implications for climate policies and international commitments (da Silva Junior et al., 2020; Pivello et al., 2021)."

 "Beyond immediate emissions, fire also influences carbon balance over time. For example, post-fire recovery critically shapes the Cerrado's long-term carbon balance (Burton et al. 2024; Gomes et al., 2020b; Hamilton et al., 2024). If vegetation fully regenerates to its pre-fire state, there is no net effect on atmospheric $CO_2$ levels over time. However, even in this scenario, fires can influence other greenhouse gases and aerosols. Alternatively, if fire activity decreases and vegetation accumulates, the landscape may shift to a net carbon sink. Conversely, if fires reduce long-term vegetation cover, the Cerrado could become a sustained carbon source, as observed globally (Burton et al., 2024)." |
| As mentioned previously, the Methods section is missing key information (e.g. that the analysis only considers papers up to 2022, or how the trend line in Figure 2 is estimated and its significance level), and in the Results section is hard to distinguish between description of papers found through the review process and discussion | We have added more details in the Methods section to address the reviewer's suggestions:

 • Research question changed to: "How compiling published material on fire emissions in areas of the Cerrado that do not explicitly include anthropogenic land uses can provide a better understanding of |

| | |
|---|---|
| (e.g. lines 163-165, 272-278, 279-282, 330-333, 375-379, 396-398, 402-405, 441-460). | the placement of these emissions in the global carbon budget?"

• Inclusionary criteria updated: "We applied four inclusionary criteria to identify relevant literature: papers had to be (1) published in peer-reviewed journals with an impact factor greater than 1; (2) encompass the Cerrado biome; (3) be published after 2003, for a two-decade period (2003-2022); and (4) be conducted in areas that do not explicitly include anthropogenic land uses, here referred to as "natural areas". We define natural areas as those covered by natural vegetation of the Cerrado, where anthropogenic land uses do not occur (e.g. pasture and agriculture). According to this criteria, 48.66% (965,783 km2) of the Cerrado is covered by natural vegetation (MapBiomas, 2022). Because papers found by this review often do not specify the land use of their study area, we have not included papers that explicitly document fire occurring in anthropized areas, as a proxy for documenting existing literature on natural areas of the Cerrado. To improve the assessment of our research question, we have also incorporated in our review papers that don't focus only on the Cerrado, but rather include it as part of the analysis."

• Sentence added to explain the period of analysis: "We then conducted the search for a two-decade period, covering research from 2003 to 2022."

[revised manuscript text omitted]

"This review demonstrates that papers fail to report on fire emissions itself, with fire dynamics and parameters used to estimate emissions in the Cerrado often being the focus of published literature." |
| They also do not discuss the mitigation potential for Brazil in LU emissions, and the | The purpose of this paper is not to discuss land use, but rather land management in the context |

| | |
|---|---|
| impact and importance of such policy changes in keeping to the 1.5ºC goal (see Roe et al., 2019 in Nature Climate Change). | of the potential of fire management in mitigating fire emissions in the Cerrado. With reference to fire management, we have added the impact and importance of fire policies in keeping to the 1.5oC goal in the Discussion section: Thus, this review indicates a critical need to develop interdisciplinary studies to bridge fire policies and fire emissions in the Cerrado. Understanding fire dynamics, including the opportunities for mitigating emissions from fire activities, is essential for recognizing fire's role in achieving global environmental and climate targets. For instance, Martin (2019) identifies United Nations Sustainable Development Goals that are related to fire and land management, including fire management, as goals 3 (good health and well-being), 13 (climate action), and 15 (life on land). These impact the 2015 Paris Agreement target to limit warming to 1.5 °C by 2100. The Paris Agreement outlines commitments for climate actions and acknowledges the importance of mitigation and removal actions, where fire management can play an important role. The 1.5 °C target is ambitious, yet achievable if great effort is put into mitigating emissions and removing carbon, with Brazil holding the highest mitigation potential in the land sector (Roe et al., 2019). Together with other countries, improved forest management – which includes fire management – in Brazil could be able to increase carbon removal by 40 $GtCO_2$ by 2050 (Roe et al., 2019).

 Climate change increasingly affects fires, and adaptation and mitigation activities are essential to limit these effects (Burton et al., 2024). Direct human impacts may offset the effects of climate change in fire worldwide (Burton et al., 2024), especially in fire-prone environments, and this is an opportunity to investigate the potential of fire management to mitigate emissions in the Cerrado, and to understand fire emissions in the biome. Pathways towards improving fire emissions in the Cerrado include connecting observational |

| | information with modeling and a better assessment and quantification of the impact of qualitative aspects in fire estimates. Examples of how this can be achieved is by valuing prescribed burning emissions and including these in fire modeling, representing fire management in land surface models, using on-site observations to assess models' utility and as input data to modeling, and incorporating non-carbon aspects of fire in fire emission estimates, such as the ecological, social and cultural aspects. These could address uncertainty and improve models' accuracy, thus providing better accounting of fire emissions in the Cerrado and worldwide." |
|---|---|
| Line 441: I believe the authors are confusing emission factors with carbon emissions. | This sentence refers to the complexity in estimating fire emissions, reflected in the different values and units reported in Table 2. Emission factors are reported in Table 1, while table 2 summarizes fire emissions in the Cerrado found in the literature, which are estimated through different methods and thus result in different values. |
| Line 327: "low fuel moisture and low flammable biomass" if there is low fuel moisture, there should be high flammability. Please clarify. | Figure 4 was edited and this sentence was excluded from the legend. |
| Line 380: "GFED relies on the study done by (...) to quantify emissions worldwide" GFED doesn't rely on Van Der Werf et al. (2017). Its fourth version is described in that paper. Moreover, "small burned areas detection derived from MODIS" seems to entail that GFED did not rely on MODIS, which is incorrect. Please clarify how small burned areas were included in the GFED dataset (which also relies on active fire information). | Thank you. These were fixed in the text and replaced by "GFED quantifies fire emissions globally, and estimations are based on MODIS burned area products and on the Carnegie–Ames–Stanford Approach (CASA) model (Van Der Werf et al., 2017). Version 4s of GFED also includes small burned area detection to improve its results, and small burned area detection in GFED4s relies on MODIS burned area product, on active fire from MODIS, and on surface reflectance observations (Van Der Werf et al., 2017)." |

| | |
|---|---|
| Line 558-559: how are fire emissions a sink of CO2? | This was not phrased correctly and 'fire emissions' was replaced by 'fire dynamics'. |
| Standardize units throughout the manuscript (e.g. Pg year-1 or Pg per year) | The manuscript has been revised to standardize units to Pg year$^{-1}$. |
| Please write biome in lowercase. | Biome is now in lowercase throughout the manuscript. |
| Line 502: "we found" should be "Van Der Werf et al. (2017) found" | "We found" replaced by "Van der Werf et al. (2017) found". |
| Authors' contribution is missing an author. | All authors were included in the sentence "All authors interpreted and analyzed data." To make this clearer, the sentence is now replaced by "RMV, CvR, CB, DIK, MC and FM interpreted and analyzed data". |
| Either "burnt area" or "burned area". Both are used in the manuscript. | "burnt area" replaced by "burned area" throughout the manuscript. |
| Figure 3 has 25 papers in the Cerrado, while the text mentions 26 (line 182). | The correct is 26 papers. This has been fixed in the new Figure 3. |
| Figure 3 and 4 could be merged into one. | Figures 3 and 4 were merged into the new Figure 3. |
| Figure captions need to be much more detailed. | Figure captions were improved as follows:

• Figure 1: Adapted PRISMA flow diagram demonstrating the systematic literature review process divided into three steps: identification of potential papers through searched terms in the Google Scholar database, and exclusion of papers based in the four criteria established for this research; screening of the papers selected and exclusion of papers with the reported reasons; and inclusion of papers in this literature review.
• Figure 2: Number of papers published per year from the 69 papers included in this literature review, from 2003 to 2022. |

|  | <li>Figure 3: Number of papers per study design and per coverage of study area in both percentage (chart) and actual numbers (data table).</li><li>Figure 4 (new figure added in response to Reviewer 2 to include the institutions that lead the papers selected for this review): Division of the institutions of the first authors from the papers reviews. The chart on the left indicates that 43 papers involve first authors from international (non-Brazilian) institutions, while 26 come from Brazilian institutions, of which 10 are within the Cerrado region. The chart on the right indicates that, from the international-led papers, 14 involve authors from Brazilian institutions, while 29 do not.</li><li>Figure 5: Variables associated with estimating fire emissions in the Cerrado found in the literature. The Cerrados's physiognomies, separated into forests, savannas and grasslands, increase in fine fuel load and decrease in fuel moisture from forests to grasslands. Microclimatic conditions also change across the physiognomies, with increasing wind speed and air temperature, and decreasing relative humidity from forests to grasslands. The Cerrado's seasonality is divided into wet and dry seasons. The wet season is characterized by high precipitation, lightning ignitions and accumulated biomass, whereas the dry season is characterized by low precipitation, anthropogenic ignitions and flammable biomass. Fuel characteristics (square boxes), climatic conditions (circle boxes) and ignition (hexagon boxes) interact (dashed lines) to determine the Cerrado's fire behavior. Two aspects of fire behavior are presented (numbers 1 and 2): 1) fire spread increases from forests to grasslands; 2) fire intensity increases in the dry season. The Cerrado's physiognomies, seasonality and fire behavior together drive the size of burned area, resulting in fire emissions (solid lines).</li> |
|  |  |

| | High-intensity fires typically consume more fuel, leading to higher combustion efficiency and more complete combustion. Combustion completeness then affects the proportion of biomass converted into carbon and released into the atmosphere, also resulting in fire emissions (solid lines). The image representing the Cerrado's physiognomies was adapted from the Brazilian Agricultural Research Corporation (Embrapa n.d.). |
| | • Figure 6: Treemap of the methodological techniques used across study areas in empirical papers. The numbers represent the number of studies of each methodological technique within each study area. The study areas Global, Tropical region, South America and Brazil are regions that include results for the Cerrado. Some papers combine different techniques and are double-counted. |
| Line 489: dos Santos et al. (2021) found that fire management reduced LDS in 3 PAs of the Cerrado, not in "areas of the Cerrado", this should be clear. | This refers to the Schmidt et al. (2018) reference. The sentence was replaced by "Fire management has reduced LDS area burned by 40-57% in the three PAs encompassed in the Cerrado-Jalapão project during the first three years (2014-2016) of implementation (Schmidt et al., 2018)." |

---

## Author Comment (AC2)

**Response to comments by referee #2 on the manuscript egusphere-2024-2348**

We, the authors, thank the editor for handling the paper and the reviewer for their comments and suggestions. We value the careful feedback provided, and we believe this is important for improving the quality of our review paper. We provide a table with detailed responses to each separate comment. According to the editor's instructions, the revised manuscript should not be prepared at this stage. Therefore, we have not included the specific line(s) or page(s) where changes were made, nor the updated figures and tables. Instead, we have provided the edited or added paragraphs and sentences to demonstrate how we have addressed the reviewer's comments.

Sincerely,

Renata Moura da Veiga (on behalf of all co-authors)

| Reviewer 2 | |
|---|---|
| **Comment** | **Authors' response** |
| The introduction provides a useful overview of the geographical, seasonal, vegetative, and emission characteristics of the Cerrado. However, it overlooks two crucial aspects: (1) The Cerrado's role in water resource availability in Brazil, as it is responsible for surface water in 8 of the 12 major Brazilian hydrographic regions, and how climate change and extreme fire events could impact the hydrological cycle. (2) The socio-biodiversity of the Cerrado, shaped by its peoples' socio-cultural relationships with nature, is highly relevant when considering the connection between ancestral knowledge and integrated fire management. I believe these points would enhance the introduction. | We have included two sentences in the introduction to contemplate these suggestions. However, we did not go too deep into these, to also contemplate the reviewer's 1 view of not focusing too much on fire dynamics. The sentences are:

(1) "Drought-heatwaves episodes and extreme fire events intensified by climate change also impact hydrological processes, including precipitation and evaporation trends, groundwater recharge and soil infiltration capacity (Klink et al., 2020; Libonati et al., 2022). This is particularly important because the Cerrado region supports aquifers that supply important hydrographic basins in the whole country (Klink et al., 2020)."

(2) "The cultural, socio-economic and ecological aspects of fire are crucial to execute and evaluate IFM activities (Myers, 2006). IFM integrates traditional knowledge and its connection with fire, and Australia is a leader in documenting these (da Veiga and |

| | Nikolakis, 2022). Measuring the social and cultural dimensions of fire presents significant challenges, and often is excluded from fire emission estimates in the Cerrado." |
|---|---|
| In the introduction, you discuss how fire and climate regulate one another and can form a positive feedback loop. However, there is no mention of the interaction between droughts and heat waves, which amplifies fire risks. Recent research highlights the importance of understanding compound drought, heatwaves, and fires, which I consider essential to this work's context. | We have included the influence of compound events in fire activity in the Cerrado in the Introduction: "Fire participates in many complex interactions in the carbon cycle, from releasing carbon to benefiting ecosystems trajectories (Hamilton et al., 2024). Fire and climate regulate one another and can be in a positive feedback loop – climate and humans can influence fire patterns, and fire can influence climate by releasing carbon (Bowman et al., 2009). In the Cerrado, higher temperatures and reduced precipitation are now more common due to climate change, which also changes its fire regimes, with fire events becoming increasingly common (Gomes et al., 2020b; Hofmann et al., 2021). The IPCC AR6 WGI/WGII (IPCC, 2021, 2022) and the UNEP "Spreading like Wildfire" report (UNEP, 2022) warn that climate change increases drought conditions, which can aggravate heatwaves, increasing the risk of fire occurrence and the intensity and frequency of extreme events, such as wildfires. This happens because the combination of extreme weather events that occur simultaneously, or compound events, can amplify their effects (Silva et al., 2024). For example, the year 2020 was marked by compound drought-heatwave episodes, which favored fire activity and the increase in burned area in the Cerrado (Libonati et al., 2022; Silva et al., 2024). Drought-heatwaves episodes and extreme fire events intensified by climate change also impact hydrological processes, including precipitation and evaporation trends, groundwater recharge and soil infiltration |

| | capacity (Klink et al., 2020; Libonati et al., 2022). This is particularly important because the Cerrado region supports aquifers that supply important hydrographic basins in the whole country (Klink et al., 2020)." |
|---|---|
| You classify the studies by location range, from global to local scales, indicating that the number of studies is higher for the Cerrado and global levels. I was curious about the spatial distribution of the institutions involved in these studies. Are they predominantly Brazilian or located in the Cerrado region? In other words, who is driving research on fire impacts in the Cerrado? | Thank you for this question, this is an interesting analysis. We have now included a paragraph to answer this in the Results section:

 • "We also observed that international (non-Brazilian) institutions drive most of the research captured by this literature review. We gathered the institution from the first author of each paper, of which 43 are international (62.3%) and 26 are Brazilian (37.7%). From the Brazilian-led papers, 10 (38.5%) are from institutions located within the Cerrado area. Also, 14 papers (32.6%) from the international-led studies involve authors from Brazilian institutions (Fig. 4), while half of the Brazilian-led studies (13 papers, 50%) include authors from international institutions. These numbers indicate that most studies in fire dynamics and emissions in the Cerrado are not led by institutions within the Cerrado region. In fact, most institutions are not even located within Brazil, with international institutions leading the studies and often not collaborating with Brazilian institutions."

 • We have also included a Pie of Pie chart (Figure 4) to demonstrate these numbers. Figure legend: Division of the institutions of the first authors from the papers reviews. The chart on the left indicates that 43 papers involve first authors from international (non-Brazilian) institutions, while 26 come from Brazilian institutions, of which 10 are within the Cerrado region. The chart on the right indicates that, from the international-led papers, 14 involve authors from Brazilian institutions, while 29 do not. |

| | We have also added an analysis of the results in the Discussion section: |
|---|---|
| | • "Additionally, we found that the majority of the papers covered in this systematic literature review is driven by non-Brazilian institutions and/or do not include authors associated with Brazilian institutions. From all the papers included, only 10 involve first authors from institutions located within the Cerrado region. This indicates an opportunity to enhance collaboration between Brazilian and non-Brazilian institutions, and even a potential to increase partnership between different regions within Brazil." |
| The sharp drop in publications in 2022 is striking. Could this reflect a shift in focus toward another biome, such as the Pantanal? A simple analysis of publication trends in other biomes could provide insight. Also, might the pandemic have affected research outputs? While I understand this is not the article's focus, the significant drop warrants more than a brief mention. | We have expanded the discussion about the 2022 drop. Although we do think it could be related to the COVID-19 pandemic, we think this requires a deeper analysis that is out of the scope of this paper.

A brief search revealed papers on fire dynamics and emissions in Pantanal and in the Amazon published in 2022. We then included the following paragraph in the Results section (Systematic Literature review process subsection):

"There is an increasing tendency in the number of papers published throughout the timeseries, but the year 2022 did not follow the growth trend shown in Fig. 2. This sharp drop in publications could indicate a gap in publications in this year or a limitation of our research method that could not capture publications in 2022. It could also indicate a shifted focus away from the Cerrado studies due to political or financial constraints to encourage scientific studies in the region, or due to a shifted focus towards other regions of Brazil. For example, papers about fire dynamics and emissions in the Pantanal and in the Amazon rainforest were published in 2022 (see Barbosa et al., 2022; Dutra et al., |

| | |
|---|---|
| | 2022; Menezes et al., 2022; Silva et al, 2022; Walker et al., 2022). In Pantanal, the main focus was the 2020 extreme fire event, when burned areas were 200% greater than the average for 2003-2020 (Barbosa et al., 2022). Papers published in 2022 related to fire dynamics and emissions in the Pantanal and in the Amazon show fire as a consequence of the compound impact of land use and climate in these regions (Barbosa et al., 2022; Silva et al., 2022; Walker et al., 2022)." |
| Your findings show that 2020 was the most critical year in terms of burned area. Is there any information on what caused this increase? Could it be related to drought and exacerbated heat, or perhaps changes in government policy or legislation? This point deserves further discussion in the text. | 2020 was a critical year in terms of burned area, but not the most critical one. We have included a paragraph in the results section, in the new subsection "Burned area and fuel characteristics", to explain the 2020 fires:

"The year 2020 was a significant year in terms of burned area in the Cerrado due to a combination of factors (Pivello et al., 2021). 2020 was a drought year in the biome, intensified by prolonged dry season and heatwave (Hofmann et al., 2021; Libonati et al., 2022; Silva et al., 2024). This compound drought-heatwave episode aggravated fire activity in the Cerrado (Libonati et al., 2022; Silva et al., 2024). Although no estimates were found correlating the compound event of 2020 with fire emissions, it is expected that the drought-heatwave episode led to increased fire emissions due to the increased fire activity and burned area that occurred in that year. Also, 2020 was critical in terms of environmental policies and legislation in Brazil, which also reflected in the Cerrado (Schmidt and Eloy, 2020). The increase in deforestation, encouraged by political discourses, and the decline in environmental legislation enforcement created a favorable setting to fire occurrence in the Cerrado. The combination of climatic conditions and the intensification of an anti-environmental discourse by the Federal government favored the occurrence and spread of fires in the Cerrado in 2020, which was also observed in |

| | |
|---|---|
| | 2021, when INPE estimated 143,342 km2 of burned area in the Cerrado." |
| You identify that only 8% of papers focused on fire management, and state that "this review captured no studies quantifying the amount of fire emissions mitigated by fire management in the Cerrado." This seems to contradict the statement that "three prominent topics identified were fire dynamics, emission estimates, and fire management". I believe adjusting the scientific question or the criteria for topic selection is necessary. | Thank you for your comment. Although 8% of papers are classified under 'fire management and policy', none of them discuss fire emissions itself within fire management and fire policy. For this reason, we discuss this topic in the sense of the influence of fire management and policy in estimating fire emissions in the Cerrado.

We have made this clearer in the manuscript by editing the first paragraph of the 'The influence of fire management and policy in estimating fire emissions in the Cerrado', which now reads:

"In synthesizing the literature on fire emission in the Cerrado, we identified 8% of papers focused on fire management and policy, all under the 'review' and 'perspective' categories. This indicates that fire management and policy are important in understanding fire dynamics in the Cerrado. Still, papers that address these do not usually bring new information based on observation or experiments but tend to synthesize or opine on existing literature. **For example, this review captured no studies quantifying the amount of fire emissions mitigated by fire management in the Cerrado, probably due to the difficulty in quantifying the social and cultural aspects of fire, which are intrinsic to fire management and policy.** Estimating the influence of humans on fire emissions is a complex task, which is also reflected in the lack of equations and algorithms to reproduce fire management strategies in land surface models. That makes sense, given all factors that need to be considered beyond quantifying the amount of GHG emitted to the atmosphere." |

| | |
|---|---|
| Additionally, while there may be no studies on integrated fire management reducing emissions in Brazil, may research outside Brazil, such as in Australia have shown this potential? Expanding this discussion would add valuable global context. | In the 'The influence of fire management and policy in estimating fire emissions in the Cerrado' section, we have expanded the discussion on the potential of fire management, especially EDS burns, in reducing emissions in other savanna countries. For this, we have included the following paragraph:

"Dos Santos et al. (2021) have shown that LDS burns have higher combustion factor, heat released, and fire intensity when compared to EDS burns. Fire management has reduced LDS area burned by 40-57% in the three PAs encompassed in the Cerrado-Jalapão project during the first three years (2014-2016) of implementation (Schmidt et al., 2018). In Canastra National Park in Brazil, areas under fire management also presented less annual area burned (Batista et al., 2018). These reaffirm the potential of management activities to reduce emissions, as shown in other savanna countries. In northern Australia, more specifically in the WALFA area (West Arnhem Land Fire Abatement), a region recognized as a reference for integrating fire studies with traditional knowledge, EDS burns emit 48% of what is emitted in the LDS (Russell-Smith et al., 2009). The WALFA project applies EDS burns to reduce LDS burns, and during its first 7 years of implementation, GHG emissions have reduced more than 37% when compared to the pre-project 10-year emissions baseline (Russell-Smith et al., 2013). Similarly, Khatun, Corbera, and Ball (2017) suggest that, in the Tanzanian miombo, EDS burns could avoid carbon emissions and enhance carbon uptake by approximately 10 tC ha-1 in a 20-year period. Studies in Mozambique and Botswana explore the potential of EDS burns to reduce emissions in southern African savannas (Russell-Smith et al., 2021)." |
| The discussion on combustion efficiency values seems underdeveloped. Is 0.94 | The MCE values are considered high and are consistent with the MCE found for other |

| considered high or low? Is it normal or anomalous? | savannahs in the world. We have made this clear in the following sentences added to the manuscript, in the subsection "Combustion efficiency, combustion completeness and emission factor" within the "Fire dynamics parameters to estimate fire emissions" section: |
|---|---|
| | "Values above 0.9 tend to characterize fires in a flaming stage, and these are predominant in the Cerrado due to the dry fine fuel that are likely to rapidly burn (Hodgson et al., 2018)." |
| | "These values are considered high and are consistent with other savannas in the world – MCE in the African and in the Australian savannas have been reported as 0.938±0.019 and 0.86–0.99, respectively (Hodgson et al., 2018)." |
| More $CO_2$ or CO affects the atmospheric carbon budget in different ways, and it would be useful to discuss air pollution and atmospheric chemistry versus greenhouse gas effects, as well as comparisons with other biomes in Brazil or other savannas globally. | • We have expanded the impacts of CO and $CO_2$ on the atmosphere in the Introduction:

"During biomass burning, a large amount of carbon gases is released to the atmosphere. These emissions are mainly in the form of carbon dioxide ($CO_2$), carbon monoxide (CO), and methane ($CH_4$) – $CO_2$ and CO combined account for 95% of the carbon emitted during biomass burning (Ward and Hardy, 1991). $CO_2$ and CO are both involved in atmospheric chemistry and the greenhouse effect in different ways. CO is recognized as a major indirect greenhouse gas, meaning that it does not absorb enough terrestrial infrared radiation to be considered a direct greenhouse gas, but it influences the concentration of other direct greenhouse gases, such as $CH_4$ and tropospheric ozone, through atmospheric chemistry (Ehhalt et al., 2001).

Savanna burning dominates the emission of CO through incomplete combustion due to limited oxygen (Ehhalt et al., 2001). Similarly, $CO_2$ is released during complete |

combustion of biomass burning (Prentice et al., 2001). $CO_2$ is a major greenhouse gas, meaning that it is crucial in absorbing and trapping infrared radiation in the atmosphere, causing the greenhouse effect. However, the increased concentration of $CO_2$ in the atmosphere has intensified the greenhouse effect and warmed the Earth in alarming amounts. Thus, understanding the emission of CO and $CO_2$ during the combustion process is important to recognize the impact of these gases in fire emissions, especially in fire-prone settings like the Cerrado. Due to their importance, the studies captured by this review often report emissions in terms of carbon released by fire, including all the carbon components emitted during biomass burning, or in terms of $CO_2$ alone, due to its impact on the greenhouse effect."

- Additionally, we have provided more details in the complete x incomplete combustion in the new subsection "Combustion efficiency, combustion completeness and emission factor" within the "Fire dynamics parameters to estimate fire emissions" section. These are also reflected in changes in Figure 5.

"The area burned, typically measured via satellite or ground surveys, is one of the primary parameters for estimating emissions (Libonati et al., 2015; Mangeon et al., 2016; Silva et al. 2021). Coupled with the available biomass for burning and its characteristics — which depend on vegetation type, density, moisture and seasonal growth patterns — these elements set the stage for potential emissions. Fire intensity, driven by conditions such as dry weather, strong winds, and fuel accumulation, influences combustion efficiency. High-intensity fires tend to consume more fuel, resulting in higher combustion efficiency and more

| | complete combustion. This reduces emissions of pollutants such as carbon monoxide and particulate matter but increases emissions of carbon dioxide. Combustion completeness further influences the amount of biomass converted to carbon and released into the atmosphere. In contrast, incomplete combustion results in higher emissions of pollutants such as particulate matter and carbon monoxide and produces pyrogenic carbon, which may persist in soils over long periods. Together, these parameters allow for the estimation of emissions based on the combination of burned area, fuel load, and combustion completeness." |
|---|---|
| I believe it is essential to list all 69 articles reviewed. This could be done as a table or supplementary material, with details such as publication year, method, and category. It is unclear whether the 69 articles are all in the reference list or if those cited throughout the text are part of this selection. | The 69 papers reviewed will be included as a table in the supplementary material. The columns included are: paper title, year of publication, authors, area of study, topic, methodological technique, study design. |
| The question posed—"How compiling published material on fire emissions in natural areas of the Cerrado can provide a better understanding of the placement of these emissions in the atmospheric carbon budget?"—is not adequately addressed or answered throughout the text. My impression is that the answer is "no," due to the lack of studies with a holistic approach. If that is indeed the case, a more in-depth discussion of this point is needed. | We have improved the research question and we have done major edits to the Discussion section and we have expanded it to include a more complete discussion of the answers to the research questions, especially the lack of holistic approach towards estimating fire emissions in the Cerrado. We have included the following:

[revised manuscript text omitted]

---

## Author Response (AR2)

**Response to comments by referee #1 on the manuscript egusphere-2024-2348**

We, the authors, thank the editor for handling the paper and appreciate the reviewer's feedback and recognize the concerns raised regarding the focus and structure of the manuscript. We have thoroughly revised the manuscript to address the reviewer's comments. We have also included an updated title and list of reviewed papers, which now encompasses papers from 2023 and 2024 as well, providing a more up-to-date list.

However, we believe the paper does address the role of fire emissions in the global carbon budget. Our review aims to explore not only the emissions themselves, but also the state of research into the underlying mechanisms that govern fire emissions in the Cerrado — specifically, how bioclimatic conditions and ecosystem functions contribute to the emissions profile. This is a critical component for understanding the Cerrado's role in the global carbon cycle.

The paper follows a call for a more integrated, interdisciplinary approach to fire science, as demonstrated by the "Fire Learning AcRoss the Earth System" (FLARE) white paper (Hamilton et al., 2024). This initiative, supported by a broad community of fire scientists, highlights the need for deeper, multi-disciplinary reviews of the carbon cycle and its drivers and clearly demonstrates that there is a demand for this type of review. Our review goes beyond simply cataloging the Cerrado's emissions; it provides a holistic exploration of their drivers and potential management strategies. We highlight key research gaps that, if addressed, could provide more comprehensive insights into how emissions change over time, what factors control them, and how fire management strategies could influence these dynamics.

We believe this broader, systems-level review is timely and valuable to the scientific community, particularly in light of the increasing recognition of the need for detailed, multi-scale assessments of fire emissions and their drivers. Additionally, this manuscript is authored primarily by Brazilian researchers, offering a perspective that is deeply embedded in local understanding and expertise.

We hope this clarification helps align the manuscript's objectives with the reviewer's expectations. We look forward to addressing the specific concerns raised and refining the paper accordingly. We provide a table with detailed responses to each separate comment in the following papers.

Sincerely,

Renata Moura da Veiga (on behalf of all co-authors)

| Reviewer 1 | |
|---|---|
| **Comment** | **Authors' response** |
| This literature review proposes to clarify the role of Cerrado's fire emissions in the global carbon budget through existing literature. To do so, the authors spend a large portion of the manuscript discussing papers that do not evaluate fire emissions but rather several aspects of fire behaviour and regime. Although knowledge on fire dynamics contribute to improved emission estimates, none of the papers listed do so | We appreciate the reviewer's feedback and acknowledge the need to maintain a strong focus on fire emissions. Our intent with this review is to assess the role of the Cerrado's fire emissions in the global carbon budget, but we also recognize that dedicated research on fire emissions in this region remains limited. Compared to other savanna ecosystems, such as Australia, where fire emissions are better quantified, the Cerrado, despite contributing significantly to global fire emissions, has received less research attention in this area. As a result, there are relatively few studies that directly estimate fire-driven carbon emissions, while many studies on fire dynamics in the Cerrado do not extend their analysis to emissions (and, in some cases, explicitly state this gap). |
| | Given this research landscape, our review takes an important step in bridging the gap between fire dynamics and emissions research. By examining how fire dynamics shape emissions and identifying where further integration of these fields could improve our understanding, we aim to encourage future studies that explicitly quantify emissions. We also highlight how insights from fire behavior and fire regimes are crucial for emissions estimation — providing necessary context on the drivers of emissions, their variability, and potential mitigation strategies. |
| | To clarify our focus, we have changed the title to better represent this broad analysis of fire |

| | |
|---|---|
| | emissions in the Cerrado. We have also refined our discussion to more clearly distinguish between studies that quantify emissions and those that provide essential context for emissions estimation. Additionally, we have made explicit recommendations on how fire dynamics research can better contribute to fire emissions assessments. We believe these refinements will strengthen the manuscript while ensuring that our review remains both focused and constructive for advancing research in this area. |
| The authors also put a lot of focus on fire management, while it is clear that there isn't enough research on fire emissions in the Cerrado to properly characterize the fire-driven carbon fluxes in the biome. I would advise the authors to focus on fire emissions and on attempting to answer the research question (which does not pertain to fire dynamics). | Fire management and policy emerged as key themes in our literature search because they play a crucial role in shaping fire regimes and, consequently, fire emissions in the Cerrado. While there is limited empirical research quantifying the direct impact of fire management on emissions in this region, we believe that discussing management strategies is essential to fully assessing the factors that influence fire-driven carbon fluxes. Understanding how management affects fire regimes provides a pathway to improving emissions estimates and identifying potential mitigation strategies.

To ensure that fire management is framed in direct relation to emissions, we have thoroughly revised the section *The influence of fire management and policy in estimating fire emissions in the Cerrado*. These revisions clarify how fire management influences emission estimates and reinforce the section's relevance to our core research question. Additionally, we have streamlined discussions on fire dynamics to maintain a stronger focus on their implications for emissions estimation. |

The authors did not clarify why they focus on "natural areas". I don't see the reason for making the distinction between natural and anthropogenic land covers in regards to fire emissions. Fire is use for a variety of purposes in the Cerrado, such as agriculture and pasture management. These occur in anthropic areas and also contribute to fire emissions. Given that around 50% of the Cerrado biome is anthropogenic land cover (Colman et al., 2024) and a considerable portion of burned areas occur in these lands (Silva et al., 2021), I would argue that, by excluding fires in anthropogenic land covers, this review will inevitably yield erroneous results as to the role of fire emissions of the Cerrado in the global carbon budget. Moreover, even if the authors proceed with this criteria of fire emissions in "natural areas", as I stated in the previous round of revisions, a keyword should be provided as all biome-wide and global studies will consider anthropogenic land use.

We appreciate the reviewer's comments and understand the concern regarding the distinction between natural and anthropogenic land covers in relation to fire emissions. Our focus on natural areas is not intended to downplay the role of agricultural and pastoral fires but to provide a clearer understanding of how fire in these landscapes interacts with biophysical and ecological processes that shape the Cerrado's carbon cycle.

There are several reasons for this approach:

- **Fire as a Biophysical Process in the Carbon Cycle** – Fires in natural Cerrado ecosystems are closely linked to climate, vegetation structure, and ecosystem dynamics. Fires in natural areas influence long-term carbon fluxes by affecting biomass accumulation, decomposition, and post-fire recovery. A focus on natural areas allows for a clearer evaluation of how fire interacts with ecosystem function, rather than being confounded by human-driven fire use.

- **Knowledge Gaps in Fire Regimes and Emissions** – While agricultural burning is better accounted for in emission inventories, natural fire regimes in the Cerrado remain understudied, particularly in terms of their contribution to the global carbon budget. Identifying the key drivers of fire emissions in natural landscapes provides a stronger basis for improving emissions estimates, understanding fire-climate feedbacks, and assessing long-term ecosystem resilience.

- **Management and Co-benefits** – Fires in natural areas play a central role in shaping

vegetation dynamics, biodiversity, and carbon storage. Understanding these processes is essential for evaluating fire management strategies and the potential co-benefits of this practice. Maintaining fire-adapted ecosystems can support carbon sequestration while also preserving biodiversity and water resources, contributing to broader climate mitigation efforts.

- **Maintaining a Focused Scope** – Expanding the review to include anthropogenic fire use would require a fundamental restructuring of the paper. Agricultural burning introduces additional complexities related to land tenure, policy, and socio-economic drivers. While these aspects are critical, they fall outside the intended scope of this review. By focusing on natural areas, we aim to provide a clearer ecological perspective on fire emissions in the Cerrado and their implications for the global carbon budget.

We recognize the need for clarity in how we define our scope. To address this, we have refined the manuscript to explicitly state that our focus is on fire emissions from natural areas, while acknowledging the role of anthropogenic fires and the importance of further research integrating both systems. We hope this clarification aligns with the reviewer's expectations.

The paragraph now reads (lines 122-132): "We applied four inclusionary criteria to identify relevant literature: papers had to be (1) published in peer-reviewed journals with an impact factor greater than 1; (2) encompass

| | the Cerrado biome; (3) be published after 2003; and (4) be conducted in areas that do not explicitly include anthropogenic land uses. Anthropogenic burning, as agricultural and pastoral, is better documented and accounted for, while natural fire regimes in the Cerrado remain understudied, particularly in terms of their contribution to the global carbon budget. Although we acknowledge the role of anthropogenic fires and the importance of further research to integrate fires in natural and anthropogenic areas to fully assess fire emissions in the Cerrado, we focus on non-anthropogenic areas, or natural areas, to provide a clearer ecological perspective on fire emissions in the Cerrado and their implications for the global carbon budget. Thus, identifying the key drivers of fire emissions in natural landscapes provides a strong basis for improving emissions estimates, understanding fire-climate feedback, and assessing long-term ecosystem resilience in the Cerrado. Because papers found by this review often do not specify the land use of their study area, we have not included papers that explicitly document fire occurring in anthropized areas." |
|---|---|
| The authors state that they aim to understand how measurements are performed, how to use different non-standardized measurements, and the available products. I would argue that this goal was not met, as it is not explained anywhere how measurements of emission factors and carbon emissions are obtained, much less a discussion on which methods/measurements are best and limitations. | We have excluded this item from the aims list. While our review does provide information on how measurements are done to estimate fire emissions in the Cerrado, we appreciate the reviewer's comment and understand that this is not one of the main goals of the paper. The paragraph was reformulated:

"This systematic literature review aims to gain a comprehensive understanding of the Cerrado's fire emissions within the global carbon budget by evaluating how fire |

| | |
|---|---|
| The authors do not list available products and/or explore in the Discussion other methods that were not found in the Cerrado literature. | parameters can inform emission estimates and mitigation strategies. Particularly, we aim to: (a) outline current emissions estimates, specially carbon, in regions that encompass the Cerrado or are limited to it; (b) describe fire dynamic factors that support these estimates; (c) understand how these estimates fit the carbon budget; (d) identify mitigation strategies in the biome and their link to fire-driven carbon fluxes; and (e) identify research gaps. This will support improvements for future fire emission estimation, provide insights into the placement of the Cerrado in the global carbon balance and assist fire policies." |
| | We show what are the main methods used to estimate emissions in the Cerrado documented in literature, and how these are performed. By synthesizing literature, we find that the main methodological techniques used in the Cerrado are models, satellite and in situ observations, with results from literature reviews also being the source of input data in studies. |
| | We provide examples of studies that measure fire dynamics and emissions and their methodologies throughout the manuscript. To improve this analysis, we have included additional information found in the reviewed papers, such as lines 508-513: "Two global emissions datasets were often used in the papers reviewed to develop and evaluate models of fire occurrence and effects, and these are GFED and Global Fire Assimilation System (GFAS). Both rely on MODIS products to estimate emissions. GFED fire emissions estimates are based on MODIS burned area products and on the Carnegie– |

| | Ames–Stanford Approach (CASA) model (Van Der Werf et al., 2017). GFAS estimates fire emissions globally by using a conversion factor that links MODIS-derived FRP observations to combustion rates, resulting in the fuel consumption, which is then combined with emissions factors to estimate fire emissions (Kaiser et al., 2012)." |
|---|---|
| It is still unclear to the reader, in the Results section, if the papers mentioned are included in the review or not. For instance, which paper gave information on how burned area and fire intensity contribute to emission estimates in lines 319-328? Only three papers are mentioned in the beginning of this paragraph: one of them is not in the literature review and the two others do not discuss or evaluate fire emissions. If none of the papers discuss this, then this information should not be in the Results section. | • We have updated Table S1, which now includes a column to contemplate the subsection of Results of each paper, entitled 'Results' subsection', to simplify the understanding of the papers that are included in the Results section, and to which subsection they are under.

• Lines 319-328 were moved to Discussion (now lines 685-694).

• The paragraphs that introduce the section 3.2 Fire dynamics parameters to estimate fire emissions were modified and now read (lines 318-327):

"Papers under 'fire dynamic parameters' encompass 46% of the studies reviewed, underscoring the importance of variables like burned area, fuel characteristics, combustion completeness, combustion efficiency and emission factor in fire emissions research. These parameters directly influence emission estimates, with their combination playing key roles in determining carbon emissions from fires. By examining these variables within the specific ecological and climatic context of the Cerrado, we gain insights into how fire behavior and emissions in this biome interact.

The prevalence of studies on fire dynamics parameters reflects the accessibility of these |

| | |
|---|---|
| | variables and indicates the importance of linking fire dynamics directly to emission, with studies often highlighting the potential applicability of their research in fire emission estimation (e.g. Libonati et al., 2015; Pereira Junior et al., 2014). This focus on fire dynamics provides some of the most current information available, yet it suggests a need for more research to correlate fire drivers to emissions. We further discuss the fire dynamics parameters found in the literature review process." |
| There are several more instances where papers that were not found in the literature review (and thus not listed in the Supplementary Material) are cited in the Results section: for example, Haas et al. (2022) - line 357; Bistinas et al. (2014) - line 368; Ichoku et al. (2018) - line 418; several in lines 484-485; Bustamante et al. (2018) - line 558. Conversely, there are many papers found in the literature review that are not mentioned in the text and their contributions to the subject remain undiscussed. | We have revised the manuscript to make sure the Results section only includes papers found in the literature review process, except when it refers to concepts, and these were published before the time period covered by this review (Carvalho Jr. et al. (1998), line 371; Ward and Hardy (1991), lines 373 and 499; DeBano et al. (1998), line 422; Wooster (2002), lines 428, 429 and 434).

While we acknowledge the value of all papers found in the literature review, not all could be included in the main manuscript. The papers referenced in the Results section highlight the main findings and key themes from the literature review process (fire dynamics, emissions, management and policy in the Cerrado). However, the full list of reviewed papers remains available for transparency and further reference in the Supplementary Material. |
| Several sections of the Results belong in Discussion as they are not stating what was found but rather an interpretation of the authors (e.g. lines 220-226, 274-277, 344- | We have reviewed the mentioned sentences and have moved them accordingly, when appropriate. |

| | |
|---|---|
| 354, 501-504, 552-559, 594-607). Likewise, information that would be relevant in Methods is still stated in the Results section (e.g. lines 227-229, 239-243). | Lines 220-226: Moved to Discussion (lines 646-653)

Lines 274-277: Moved to Discussion (lines 657-660)

Lines 344-354: Deleted.

Lines 501-504: Deleted.

Lines 552-559: Moved to Discussion (lines 661-669)

Lines 594-607: Moved to Discussion (lines 707-721)

Lines 227-229: Moved to Methods (lines 135-137)

Lines 239-243: Moved to Methods (lines 192-197) |
| Lastly, it would help a lot to know which papers are being analysed and belong to each subsection of Results. | The supplementary Table S1 provides a column entitled 'topics', which indicates the topics under which the papers were classified. We have updated Table S1, which now includes a column to contemplate the subsection of Results of each paper, entitled 'Results' subsection'. |
| The authors could easily employ simple statistical methods that would add significance to their results. Indeed, in the previous version of the manuscript there was a trend line in Figure 2 that, albeit lacking information on how it was computed and its parameters, added statistical significance to the clear upward trend in papers over the last 2 decades. However, the authors removed this analysis but still state in the text that: "There is an increasing tendency in the number of papers (...)" (lines 218-219). | We have included again the trendline in Figure 2, together with its p-value (p=0.005). The trendline indicates an increase in papers published throughout the time series.

Lines 218-219 (now 229-230) now reads: "There is a statistically significant increasing trend ($p \leq 0.005$) in the number of papers published throughout the time series, with a sharp drop in publications in the year 2022". |

| | |
|---|---|
| Please use en dashes for ranges (e.g. lines 381, 411, 519, 591, 592) | En dashes added to all ranges throughout the manuscript. |
| Line 50: this is incorrect. Neither Gomes et al. (2020b) nor Hofmann et al. (2021) show that fires are increasing in the Cerrado. Recent papers suggest otherwise (e.g. Andela et al., 2017). | The sentence was reformulated and now reads (lines 69-70): Fire events are also becoming more frequent and intense (Gomes et al., 2024; Oliveira et al., 2022; Pivello et al., 2021)". |
| Line 222: please see Pereira et al. (2024) "Changes, trends, and gaps in research dynamics after the megafires in the Pantanal". They do refer to a shift in the academic community's attention pre and post the 2020 megafires. | Thank you for the suggestion. We believe referencing the indicated paper strengthens our argument of a shifted focus away from the Cerrado studies and towards other regions of Brazil. We included the following sentence in Lines 649-650: "Pereira et al. (2024) indicate an increase in papers published about fires in the Pantanal after the 2020 megafire in the biome." |
| Lines 329-330: the prevalence of studies in fire dynamics do not show a "gap in linking fire dynamics directly to emissions" as they do not propose to do so. | The sentence was improved and now reads (lines 323-325): "The prevalence of studies on fire dynamics parameters reflects the accessibility of these variables and indicates the importance of linking fire dynamics directly to emission, with studies often highlighting the potential applicability of their research in fire emission estimation (e.g. Libonati et al., 2015; Pereira Junior et al., 2014)." |
| Line 336: Libonati et al. (2015) do not consider plant complexity. They do discuss that regional products better reflect different vegetation types. | The sentence was reworded and now reads (lines 338-339): "Libonati et al. (2015) developed a regional algorithm using MODIS data to increase the accuracy of estimations of burned area in the Cerrado, demonstrating that regional products more accurately capture vegetation diversity." |

| | |
|---|---|
| Line 338: Silva et al. (2021) do not propose to divide Cerrado into 19 ecoregions based on fire parameters. They do use the ecoregional framework and evaluate fire. | The sentence was reworded and now reads (lines 339-342): "Also, to capture the variety of fire dynamics throughout the biome, Silva et al. (2021) map fire characteristics for the 19 ecoregions of the Cerrado, including the patterns and trends of burned area using MODIS data. Results show a great variation of size of burned area in the Cerrado, with large areas detected in the boundaries with other biomes (Silva et al., 2021)." |
| Lines 340-343: this is incorrect. Looking into INPE's Queimadas website (https://terrabrasilis.dpi.inpe.br/queimadas/aq1km/) the value for 2020 is 138,540 km2 and 74,085 km2 for 2009, different from the values presented by the authors. 2020 was also not a critical term in terms of wildfire in Brazil, as easily confirmed in INPE's Queimadas website, either using active fires or burned area. It was, however, a critical year for Pantanal. | INPE Queimadas is constantly being updated and validated, and changes in the values may occur over time. This is advised on the website (https://terrabrasilis.dpi.inpe.br/queimadas/aq1km/): "The product is constantly updated and validated, so changes may occur as new versions are released. For this reason, we do not have the data available for download or other forms of analysis. If necessary, contact the INPE Queimadas team to facilitate access to the data."

 This means that, between the writing and the revision of the manuscript, INPE Queimadas was updated, and the values were slightly changed. We have updated the values in the manuscript to match the most recent values shown on the website: 2009 now presents the updated value of 74,085 km2. 2020 was replaced for 2007, the second largest year in terms of total burned area in Brazil between 2003-2024. |
| Lines 344-345: Pivello et al. (2021) do not show that 2020 was a significant year in terms of burned area in the Cerrado. They do | This paragraph was deleted. |

| | |
|---|---|
| say that it was an extreme year for Amazon and Pantanal | |
| Lines 345-347: this is incorrect. Neither Libonati et al. (2022) nor Hofmann et al. (2021) show that 2020 was a drought year in the Cerrado, as none of the studies even consider 2020 in their historical time series. And dos Santos et al. (2024) does not discuss Cerrado. | This paragraph was deleted. |
| Lines 347-349: I'm fairly certain that the authors are confusing Cerrado with the Pantanal. | The authors' list includes Brazilian scientists with extensive experience working in the Cerrado and are well aware of the differences between the Cerrado and the Pantanal. The analysis and conclusions presented in the manuscript are specific to the Cerrado and based on a thorough understanding of its distinct characteristics and dynamics.

We have changed the year that represents the critical year in terms of wildfires in Brazil: 2020 was replaced by 2007, since 2007 presents the second largest total burned area in the country between 2003-2024, according to INPE Queimadas database (https://terrabrasilis.dpi.inpe.br/queimadas/aq1 km/). 2009 was maintained as an example of a year with low overall rate of burned area in Brazil: from INPE Queimadas database, 2009 was the second year with least total burned area in the time series (2003-2024). |
| Lines 366-367: reference?

Lines 409-410: reference?

Lines 412-414: reference?

Lines 548-551: reference? | • Lines 366-367 (now lines 362-367): paragraph reformulated, and new references added.

• Lines 409-410, 412-414 (now lines 419-425): paragraph edited, and references |

| | included. Now reads: "Fire behavior is limited by fuel characteristics and availability, and microclimate conditions. Fire behavior is often analyzed in terms of fire intensity (Gomes et al., 2020a; Silva et al., 2021), fire spread (Gomes et al., 2020a), heat released (Gomes et al., 2020a), fuel consumption (Andela et al., 2016), and fire return interval (Gomes et al., 2020b; Pereira Junior et al., 2014). For the Cerrado, this means that fire intensity follows a seasonal pattern, increasing in the dry months (Silva et al., 2021), and that it is also highly influenced by the vegetation type, increasing from forests to savannas and grasslands, where fine fuel consumption is higher (Gomes et al., 2020a). Silva et al. (2021) indicates higher values of fire intensity at the end of the dry season in the Cerrado,when fuel moisture is lower and fuel availability for burning is higher."

• Lines 548-551: Paragraph deleted. |
|---|---|
| Lines 382-383: these acronyms were already defined in the Introduction (e.g. line 47 for LDS). | LDS acronym removed from the parentheses. EDS appears for the first time in this sentence and, therefore, the acronym was not removed. |
| Lines 417-419: this is incorrect, as I mentioned in the previous round of revisions. | This paragraph was rewritten and now reads (lines 427-434): "Fire intensity can also be measured through the fire radiative power (FRP). FRP is the instantaneous amount of energy released by fire in the combustion process (Wooster, 2002). FRP often derives from MODIS data, and it relates to the intensity of fire and to the amount of biomass being consumed (Wooster, 2002). Although FRP has been used to provide estimates of fire |

| | intensity, Sperling et al. (2020) states that FRP from MODIS is underestimated for the Cerrado. Through FRP, Silva et al. (2021) estimates fire intensity in the Cerrado, with high values (FRP > 63.7 MW) found in the border with other biomes. Continuous observations of FRP can be integrated over time, resulting in the Fire Radiative Energy (FRE) (Van der Werf et al., 2017). FRE represents the total amount of energy released by fire during the combustion process, and it relates to the total amount of biomass consumed by fire, being directly related to fire emissions (Wooster, 2002)." |
|---|---|
| Figure 5: very confusing figure. It doesn't help in understanding how each of these parameters contribute to the end goal of estimating fire emissions. At best, this figure should be discussed in section 3.3. | This figure is essential for illustrating the key controls on fire emissions in the Cerrado and should remain in its current placement. The figure visually synthesizes how various factors—such as fuel availability, vegetation characteristics, and meteorology—interact to shape fire properties (burnt area and fire intensity) and ultimately determine emissions. Given that our review focuses on fire emissions, this conceptual framework provides necessary context by clarifying where emissions originate and what drives their variability. |
| Line 511: this is incorrect. Gomes et al. (2020a) does not use the BEFIRE model. And these are not estimations of carbon emissions from fires in Cerrado. These parameters only show that the amount of carbon released depends on land cover type. The study does not quantify emissions. | Gomes et al. (2020a) models fire-associated emissions, more specifically carbon emissions associated with fine fuel consumption, as stated in page 2 of their research "In this study, we compiled literature data from 65 prescribed fire experiments in the grasslands, savannas, and forests of the Brazilian Cerrado to determine (1) how aspects of fire behaviour (fire spread rate, fire intensity, and heat released) and fire-associated emissions (fine |

fuel consumption, combustion factor, and carbon emissions associated with fine fuel consumption) vary according to the vegetation type (grassland, savanna, and forest) and (2) the relative importance of vegetation and microclimatic factors in determining the fire behaviour and fire- associated emissions."

They use the same methods and equations as the BEFIRE model (Gomes et al., 2020b) to estimate fire-associated emissions, as described in Gomes et al. (2020b) and in the Supplementary Materials (Table S2) of Gomes et al. (2020a).

The paragraph in Line 511 was rewritten to provide a clearer explanation, and now reads (lines 546-554):

"Gomes et al. (2020a) modelled carbon emissions associated with fine fuel consumption, finding 0.230 kg m$^{-2}$ for grassland, 0.210 kg m$^{-2}$ for savanna, and 0.053 kg m$^{-2}$ for forests, and concluding that fine fuel load was the main predictor of the amount of carbon released through fire. When considering different scenarios (moderate, medium, and extreme) for fine fuel available for burning, wind speed, and vapor pressure deficit using the BEFIRE (Behavior and Effect of Fire) model, Gomes et al. (2020b) showed that carbon emissions from fine fuel consumption increased with the intensity of the scenario (0.19 kg m$^{-2}$ for moderate, 0.23 kg m$^{-2}$ for medium, and 0.26 kg m$^{-2}$ for extreme). Because the model only considers fine fuel load, which is more abundant in grasslands due to the presence of grasses, carbon emissions decrease with the increase of woody biomass. These simulations confirm

| | that fire-associated emissions depend on the vegetation type (Gomes et al., 2020a, 2020b)." |
|---|---|
| Line 570: "That makes sense" is not adequate scientific writing. | Sentence replaced by (lines 592-593): "This complexity emphasizes all factors that need to be considered beyond quantifying the amount of carbon emitted to the atmosphere." |
| Line 583: "Despite the relevance of fire management to fire emission", reference? | We have included papers from 2023 and 2024, to provide an up-to-date systematic review. This sentence and paragraph was changed and now reads (lines 610-618): " Despite the increasing recognition of fire's importance to the Cerrado and the subsequent expansion of fire management operations, as well as the global relevance of fire management in reducing emissions (Moura et al., 2019), this review identified only one study that quantifies the reduction in fire carbon emissions achieved through fire management in the Cerrado. Franke et al. (2024) show an emission abatement of 26,677 tCO$_2$e year$^{-1}$ (2014 – 2019) in specific protected areas, and a reduction potential of more than 1,085 tCO$_2$e year$^{-1}$ (2014 – 2019) when the result is scaled-up to all protected areas in the Cerrado. The reduction in emissions from prescribed burning is due to lower fuel consumption and combustion factors of early dry season fires when compared to mid/late dry season fires (Franke et al., 2024). These values are considered conservative due to analyzing mainly fine fuels, and Franke et al. (2024) argue that estimations could be improved by using high-resolution data that would allow the identification of small-scala fires." |

| | |
|---|---|
| Lines 610-611: it would be interesting to see how many papers actually evaluate emissions and their trends throughout the study period. | We have included in the Results section, 'Systematic literature review process' subsection, a paragraph showing the number of papers that evaluate fire emissions and their trends (lines 293-298): "Of the 43 papers that report fire emissions, 23 focus only on fire emissions, while 18 analyze fire emissions and fire dynamics parameters, 1 focuses on fire emissions and fire management, and 1 on the three topics. From the 23 papers exclusive to fire emissions, only 2 are restricted to the Cerrado, one focusing on net $CO_2$ (Gomes et al., 2024), and the other on fine particulate matter (Mataveli et al., 2019). The remaining papers include the Cerrado region but are not limited to it (9 provide a global analysis, 5 relate to the Tropical region, 4 to South America and 3 to Brazil). These numbers demonstrate the potential to expand the study of emissions from fire in the Cerrado." |
| Figure 6: this should be results, as it is characterizations of the papers found in the review process. | Figure 6, now Figure 5, was moved to the Results section, 'Systematic literature review process' subsection. |
| Line 653: remove the "Therefore". | 'Therefore' deleted and sentence edited (now line 669). |
| Lines 658-659: again, how are fire dynamics a sink of CO2? | Paragraph replaced by (lines 670-677): "Examining fire carbon emissions reveals that local emissions reflect the global carbon budget. A key factor in carbon balance analysis is vegetation regrowth, since a significant portion of the $CO_2$ emitted by fire is sequestered during post-fire biomass recovery (Andreae, 2019; Van Der Werf et al., 2017; Gomes et al., 2024). This literature review identified only one study that quantifies the net $CO_2$ emissions from the |

| | Cerrado fires from 1985–2020 (Gomes et al., 2024). Vegetation regrowth removed 63.5% of the $CO_2$ emitted, indicating that fire in the Cerrado has been a source of carbon to the atmosphere in recent decades (Gomes et al., 2024). For a shorter time series (2015–2018), Oliveira et al. (2021) also found the Cerrado fires to be a net emitter of $CO_2$. Further research is needed to enhance the understanding of the long-term carbon balance of Cerrado fires. This literature review contributes by providing an overview of published studies on fire emissions in the region." |
|---|---|
| Table 3: please clarify what is "Fire culture" and how it relates to burned area. | More details on "fire culture" are now provided on lines 680-685: "This indicates that estimating fire emissions requires a holistic approach. For example, including the perspectives of fire culture, ecology and policy within emissions is essential given the importance of fire to the biome. Fire culture refers to the interaction between humans and fire, encompassing the factors that drive societies to use it (see Pivello et al., 2021). The use of fire, shaped by cultural traditions and socioeconomic conditions, can influence the extent of burned areas and the resulting fire emissions. Traditional communities, for instance, may occasionally use fire on a small scale (Pivello et al., 2021)." |
| Lines 680-684: repetition. Same information in lines 498-504. | Lines 680-684 deleted. |
| Line 710: replace on-site with in-situ. | On-site replaced with in situ throughout the text. |

| Line 711: "incorporating non-carbon aspects of fire in fire emissions", please clarify. | The non-carbon aspects of fire are those cited in the end of that same sentence "such as the ecological, social and cultural aspects". To improve the understanding, we have rephrased the sentence, which now reads (lines 733-737): "Examples of how these can be achieved are by valuing prescribed burning emissions and including these in fire modeling, representing fire management in land surface models, using in situ observations to assess models' utility and as input data to modeling, and incorporating the ecological, social and cultural aspects of fire in fire emission estimates. These could address uncertainty and improve models' accuracy, thus providing better accounting of fire emissions in the Cerrado and worldwide." |
| --- | --- |

---

## Author Response (AR3)

**Response to comments by referee #1 on the manuscript egusphere-2024-2348**

We, the authors, thank the editor for handling the paper and the reviewer for their comments and suggestions. We value the careful feedback provided, and we believe this is important for improving the quality of our review paper. We provide a table with detailed responses to each separate comment. We hope that the revisions align the manuscript's objectives with the reviewer's expectations.

Sincerely,

Renata Moura da Veiga (on behalf of all co-authors)

| Reviewer's comments | Answers |
|---|---|
| I think the Abstract would benefit with more detail and the key take-away messages. For instance, it is not clear that, amongst the papers that evaluated fire emissions, most were on a global or continental scale and there were only 2 papers that explicitly analysed fire emissions in the Cerrado. This severe lack of literature on the issue (especially compared to other ecosystems, as the authors reinforce), is one of the key take-aways of the paper in my opinion. Could also mention that most of these studies were conducted by international teams and a considerable number did not include Brazilian authors nor institutions based in the Cerrado biome. | We have fully revised the Abstract to include the suggestions made by the reviewer. The Abstract now reads:

 "Estimating fire emissions in the Brazilian Cerrado requires integrating fire parameters, mitigation strategies and policies. Despite the Cerrado's significant contribution to global fire emissions, research in this area is still overlooked when compared to other savanna ecosystems. Here, we provide a comprehensive understanding of the Cerrado's fire emissions within the global carbon budget by examining how fire dynamics, management and policy shape emissions. We systematically reviewed 77 papers, of which 57% address fire dynamics, management and policy. While these are key to providing a holistic understanding of fire emissions, linking them to estimates is challenging, especially due to the difficulty in valuing the qualitative aspects of fire. This review only identified two papers that explicitly analyze fire emissions in the Cerrado, and found that 17% of papers are led by institutions located within the Cerrado biome area. These numbers reinforce the urgent need for further investigation into the topic. Most papers employ different methods to achieve their results. Evidence suggests growing |

| | |
|---|---|
| | interest in fire emissions in the Cerrado, reflected in the rising number of studies over the years. More research is required to provide a more comprehensive understanding of fire emission in the Cerrado, understand fire dynamics and emissions, and identify potential mitigation measures that could help reduce the Cerrado's contribution to the global carbon budget. This could be achieved by better accounting of emission parameters across the Cerrado's vegetation types and fire regimes, and by including fire management representation in land surface models and using observational data to constrain and assess their utility." |
| Line 12: I suggest changing "countries" to "ecosystems". | "Countries" changed to "ecosystems" |
| Line 12: I think it's missing a phrase stating what this study is proposing to do (e.g. "Here, we propose to bridge this gap by (…)"), to precede results on the following sentence "Of 77 systematically reviewed papers (…)". | Sentence added: "**Here, we provide a comprehensive understanding of the Cerrado's fire emissions within the global carbon budget by examining how fire parameters guide emission estimates and mitigation strategies**. We systematically reviewed 77 papers, of which 57% address fire dynamics, management and policy." |
| Line 12: "papers" is duplicated. | "Papers" deleted. |
| Line 12: Is the 54% correct? Lines 290-291 state 46 papers for "fire dynamics" and 12 for "management and policy" (that is, 58 total for both categories). If some of these are double counted, then this number should figure somewhere in the manuscript for clarification. | Thank you for your comment. It is in fact 57% (58 papers out of 101, due to double counts). This is made clearer in the text in lines 273-277: "Of the 77 papers reviewed, 46 relate to fire dynamics parameters used to estimate emissions, 43 report the amounts of fire emissions, and 12 report fire management and policy. It's worth noting that 24 papers are related to more than one topic. These numbers indicate that most papers are not related to reporting emissions but provide information to support the understanding and estimation of fire emissions – **57% (double counts included) of papers address fire dynamics, management and policy.**" |
| Line 13: I suggest "While these are key to provide a holistic (…)". | "Drivers" deleted, and the sentence now reads "While these are key to provide"... |

| | |
|---|---|
| Line 16: "Methodological techniques" seems redundant (also on the legend of Figure 5). I suggest "Most papers employ different methods (…)". | Sentence in the abstract changed to "most papers employ different methods" ... "Techniques" deleted from Figure 5 (line 308 and figure legend) |
| Line 18: I suggest rephrasing to "More research is required to understand fire dynamics and emissions in the Cerrado and identify potential mitigation measures (…)". | Sentence rephrased: "More research is required to understand fire dynamics and emissions in the Cerrado, and identify potential mitigation measures that could help reduce the Cerrado's contribution to the global carbon budget." |
| Line 19-21: While I agree that land surface models would benefit from including fire management, I'd say it's more urgent that the scientific community works to properly quantify emission factors for Cerrado's vegetation types and across fire types (more/less intense fires, EDS/LDS fires, etc). | Sentence updated: "This could be achieved by better accounting of emission parameters across the Cerrado's vegetation types and fire regimes, and by including fire management representation in land surface models and using observational data to constrain and assess their utility." |
| I think the Introduction is quite big and could be slightly summarized and re-organized. I propose to relocate the following paragraphs: paragraphs from lines 55 to 77 could follow the first paragraph (lines 23-30), characterizing the Cerrado biome and fire activity. Then, keep paragraphs from lines 31-54, discussing fire emissions in Cerrado. And finally, paragraphs from lines 78-104. | We appreciate your comment. We have reorganized the paragraphs according to the reviewer's suggestion. |
| Line 24: I suggest "Around 49% of Cerrado (965,783 km2) is covered (…)". | "48.66%" changed to "around 49%" |
| Line 37: "However" seems misplaced. There is no contradiction. | "However" deleted. |
| Line 38: This phrase is confusing as is. I suggest something along the lines of "Thus, understanding the contribution of each of these greenhouse gases, especially CO2, in fire emissions is essential, especially in fire-prone ecosystems such as Cerrado.". | Sentence rewritten: "Thus, understanding the contribution of each of these gases, especially CO2, in fire emissions is essential, particularly in fire-prone settings such as the Cerrado. |
| Line 40: I suggest rephrasing this sentence, deleting "immediate emissions" and "fire participates". Something along the lines of "Beyond emissions, fire interacts with several components of the carbon cycle, shaping complex interactions and carbon | Sentence rephrased: "Beyond emissions, fire interacts with several components of the carbon cycle, shaping complex processes over time." |

| | |
|---|---|
| balance over time.". | |
| Lines 42-44: Needs a reference. | We have included the reference: Bond WJ, Woodward FI, Midgley GF (2004) The global distribution of ecosystems in a world without fire. New Phytologist, 165(2):525-37. https://doi.org/10.1111/j.1469-8137.2004.01252.x |
| Line 70: Define IPCC. | Definition added for IPCC and UNEP: "Working Groups I and II of the Sixth Assessment Report of the Intergovernmental Panel on Climate Change (IPCC AR6 WGI/WGII; IPCC, 2021, 2022) and the United Nations Environment Programme "Spreading like Wildfire" report (UNEP, 2022) |
| Line 78: As it's the first time this term is used in the article, and because it has not been said yet that this is one of the categories identified in the study, it is not clear what "fire dynamic parameters" is. I propose changing to "understanding fire dynamics provides grounding (…)". | "Fire dynamic parameters" changed to "fire dynamic" |
| Lines 80-82: Is this not a result? | Sentence deleted and paragraphs joined together: "In this context, understanding fire dynamics provides grounding for assessing fire emissions in the Cerrado, and the interaction between these is essential for uncovering the factors that influence the Cerrado's role in the global carbon budget and the broader implications for national and international policy. Linking fire dynamics to estimated emissions also guides mitigation by identifying aspects for potential intervention"... |
| Line 80: I suggest changing "cycle" to "budget". | "Carbon cycle" replaced by "Carbon budget" |
| Line 84: GHG is not defined. | Definition added: "greenhouse gases (GHG)" |
| Lines 87-88: Reference? | Reference added: Griscom et al., 2020 |
| Lines 89-91: I believe this belongs in Discussion. | Sentence moved to Discussion: lines 529-641. |

| | |
|---|---|
| Line 94: I suggest "what are the parameters used (…)". | "Fire dynamic" deleted and sentence now reads: "what are the parameters used"... |
| Line 95: You mean "pyrogenic carbon"? If so, this term should be employed earlier in the text. | Thank you for pointing this out. This sentence has been edited and now reads: "**Since carbon is a major contributor to atmospheric greenhouse gas levels**, this systematic review"... |
| Line 101: I suggest "describe fire parameters". | Aim (b) changed from "describe fire dynamic factors that support these estimates" to "describe fire parameters that support these estimates" |
| I think this section presents my only remaining disagreement with the authors: the use of "natural areas". I disagree with the authors that "A focus on natural areas allows for a clearer evaluation of how fire interacts with ecosystem function, rather than being confounded by human-driven fire use.", as fires in natural areas of the Cerrado are still, in their vast majority, human-driven (e.g. see Arruda et al. 2024: "Natural vegetation was the most affected, primarily due to human-driven ignition during the dry season").

 Nevertheless, I completely agree with the point the authors make in "Knowledge Gaps in Fire Regimes and Emissions" and add that it should be included in the Discussion. However, I don't think it justifies the use of "natural areas" in the study.

 I don't think that including anthropogenic burning would change the scope of the article, as the study already considers anthropogenic burning. As mentioned before, ignitions in the Cerrado are overwhelmingly human and, as such, any discussion of fire and fire emissions in the biome will consider anthropogenic-driven fires, unless it specifically evaluates lightning-induced fires or "natural fire regimes". Additionally, most papers found in the review process include burning in anthropogenic land covers and discuss the human components of fire. As there is no keyword in the review process for this, this is not explicitly accounted for throughout the manuscript. Moreover, there is no other | We understand the reviewer's concern, and appreciate the discussion since we believe this has improved the paper.

 "Natural areas" indeed was not included as a keyword search. Thus, we reconsidered this methodological decision, and have clarified what we mean by fires in natural vegetation in the context of our review paper, stating that we did not include papers that explicitly use fire for anthropogenic land uses. The new paragraph and argument for this decision are in lines 122-129:

 "We applied four inclusionary criteria to identify relevant literature: papers had to be (1) published in peer-reviewed journals with an impact factor greater than 1; (2) encompass the Cerrado biome; (3) be published after 2003; and (4) be conducted in areas that do not explicitly include anthropogenic land uses. Although we acknowledge the role of anthropogenic fires and the importance of further research to integrate these to fully assess fire emissions in the Cerrado, we focus on fires that are not explicitly used for anthropogenic land uses – as land clearing for agriculture implementation – to provide a clearer ecological perspective on fire emissions in the Cerrado and their implications for the global carbon budget. Thus, identifying the key drivers of fire emissions in the Cerrado's landscapes provides a strong basis for improving emissions estimates, understanding fire-climate feedback, and |

| | |
|---|---|
| mention of "natural areas" in Results or Discussion (except for the repetition of the research question). As such, I don't see the need to make this distinction, when this is not reflected in the Methods, Results and Discussion. | assessing long-term ecosystem resilience in the Cerrado."

The research question and PRISMA diagram were also updated accordingly. |
| Lines 139-140: I wouldn't say "there is greater certainty", it's the yearly availability of these products that start in 2003.

Lines 192-197: This information should come earlier in the Methods section, maybe after line 142. | Sentence changed to: "with full-year data available starting in 2003"

Sentence moved to lines 143-149. Paragraph now reads: "The criteria led to the initial screening of 109 papers. Although we used keywords to conduct our review, the searches still returned papers not in English, or that did not mention fire emissions. 32 papers were excluded due to being duplicates, not in English, or not mentioning fire emissions. Review and perspective papers were included in this systematic literature review to contribute to a more complete analysis of fire emissions in the Cerrado. **Review and perspective papers analyze previously published studies by evaluating existing literature (review) or expressing opinions on a specific topic(perspective) while empirical studies provide new information based on observation or experiments. Although they do not focus on bringing original research, they supply the current knowledge of a specific topic and highlight pertinent published literature (Cronin et al., 2008).** We full-text screened the remaining 77 papers to confirm they met all the eligibility criteria. Figure 1 demonstrates the systematic literature review process through the PRISMA diagram." |
| Lines 192-197: Please clarify the different between Review and Perspective papers for the reader. | Thank you for your comment. The definition is already included in lines 144-146, and we have made these clearer: "Review and perspective papers analyze previously published studies by evaluating existing literature (review) or expressing opinions on a specific topic (perspective)" |
| Line 203: I suggest just using "fire | We have kept the term "fire dynamic |

| | |
|---|---|
| dynamics" instead of "fire dynamics parameters". | parameters" in this line and throughout the text. We believe this term better contemplates the discussions made in the manuscript, since we not only discuss the overall fire behaviour and how fire interacts with the environment and climate (fire dynamics), but also specific variables that describe aspects of fire dynamics (such as fire intensity, fuel load, and so on). |
| Line 203: I think the authors meant "research". | "topic of search" replaced by "topic of research" |
| Line 229: How was the trend estimated (linear regression, Mann-Kendall, etc)? This information should be added to Methods. | Sentence added to lines 139-140: " We evaluate the trend in the number of papers published over time using linear regression" |
| Line 237: There is no contradiction, as Mistry et al. (2019) also extrapolates from a smaller region to a larger. | "Conversely" changed to "Similarly" |
| Figure 4: Please upload with higher resolution. | Figure with higher resolution uploaded. |
| Line 291: I suggest rephrasing to "(…) policy. It's worth noting that 24 papers are related to more than one topic.". | "The total does not round up to 77 because 24 papers are related to more than one topic" replaced by "It's worth noting that 24 papers are related to more than one topic" |
| Lines 298-299: This sentence belongs in the Discussion. | Sentence moved to Discussion (now lines 695-696). |
| Line 301: Please clarify what "modelling" means in this context. Statistical modelling, process-based models? | We have now defined what we mean by modelling in the sentence: "In this study, we discuss 'models' in terms of the qualitative and quantitative characterizations of components within a system and their interactions (IPBES, 2016)." |
| Line 313-314: I think this sentence belongs in the Discussion and Conclusion. | This sentence was included as the first sentence of the Discussion. This sentence was already in the Conclusion in LINES 772-774 (now 753-755):
"Based on our knowledge and search criteria, this is the first systematic literature |

| | review to provide an integrated understanding of fire emissions in the Cerrado, where fire dynamics, management and policy emerge as crucial for estimating fire emissions". |
|---|---|
| Line 325: "emissions" | Corrected. |
| Line 379: Use the acronym previously defined for modified combustion efficiency. | "modified combustion efficiency" replaced by MCE |
| Line 382: Use the acronym previously defined for late dry season fires. | "Late dry season" replaced by LDS |
| Line 384: The values are so similar it is relevant to show the uncertainty range if there is one. | Vernooij et al. (2021) do not provide a uncertainty range. |
| Lines 385-386: It is said that the values are underestimated, and the following sentence says these are high. Seems contradictory. | Sentence rephrased: "These values are consistent with other savannas in the world" |
| Line 389: I suggest "(…) estimating emissions (…)". | "estimating fire emissions" replaced by "estimating emissions" |
| Line 428: This seems to entail that the previous studies did not use FRP as a measure of fire intensity, which is not the case. | We have deleted the word "also" as an attempt to remove this interpretation. The sentence now reads: "Fire intensity can also be measured through the fire radiative power (FRP)." |
| Lines 433-435: If FRE is not used in any study I don't see why its definition is needed here. | Sentence deleted. |
| Lines 448-452: I think this belongs to the Discussion. | We appreciate your comment. Although we agree that these sentences could fit the Discussion section, we have kept them where they were. We believe they introduce Figure 6, which follows this paragraph and summarizes all the predominant factors associated with fire emissions in the Cerrado mentioned in this section. |
| Figure 6: I now agree with the authors that the figure is essential to report findings. However, I still have reservations about the arrows, namely those going to Burned Area and Combustion completeness and then Fire emissions. You found in the literature that | We have modified Figure 6 to include the elements discussed in the text: combustion efficiency, FRP and emission factor. Together with burned area and combustion completeness, these now result in fire emissions. The legend was also modified |

| | |
|---|---|
| there are other parameters (e.g. FRP; emission factors) used to estimate Fire emissions. Why are they not considered here? | accordingly. We believe these changes address the reviewer's concerns, and make the Figure less confusing. |
| Legend of Figure 6: I don't think it's "fire spreads from forests to grasslands" but rather "fire spread increases from forests to grasslands". | Legend modified: "Figure 6: Variables associated with estimating fire emissions in the Cerrado found in the literature. The Cerrados's physiognomies, separated into forests, savannas and grasslands, increase in fine fuel load and decrease in fuel moisture from forests to grasslands. Microclimatic conditions also change across the physiognomies, with increasing wind speed and air temperature, and decreasing relative humidity from forests to grasslands. The Cerrado's seasonality is divided into wet and dry seasons. The wet season is characterized by high precipitation, lightning ignitions and accumulated biomass, whereas the dry season is characterized by low precipitation, anthropogenic ignitions and flammable biomass. Fuel characteristics (square boxes), climatic conditions (circle boxes) and ignition (hexagon boxes) interact (dashed lines) to determine the Cerrado's fire behavior. Two aspects of fire behavior are presented (numbers 1 and 2): 1) fire spread increases from forests to grasslands; 2) fire intensity increases in the dry season. The Cerrado's physiognomies, seasonality and fire behavior together (red solid square) interact to determine size of burned area, combustion completeness, combustion efficiency, emission factor and FRP. These (red dashed line) drive the resultant fire emissions (red dotted line). The image representing the Cerrado's physiognomies was adapted from the Brazilian Agricultural Research Corporation (Embrapa, 2024)." |
| Line 498: Is the 43% correct? Isn't it 43 papers out of 77? | 43% is correct. It is 43 out of 101, given that 24 papers belong to more than one topic. This is made clearer now: "43% (43 papers, due to papers double counted) |
| Line 520: Acronym already defined previously. | Definition in parentheses "(Southern Hemisphere South America)" removed. |

| | |
|---|---|
| Lines 526-530: Seems Discussion to me. | Moved to Discussion: lines 634-639. |
| Table 2: I think it would be interesting a discussion on the similarity/disparity of the values found in the literature, and how they compare to broader estimates (from savannas in South America, for example). | We appreciate your comment and agree that a comparison with broader estimates could enrich the paper. However, we chose, in this section, to focus on studies explicitly dedicated to estimating carbon emissions from fires in the Cerrado. Expanding the discussion to include broader-scale estimates would, in our view, require a more extensive analysis. We believe that maintaining a focused approach allows for a more in-depth assessment of the Cerrado-specific dynamics and uncertainties. |
| Table 2: Please clarify that Gomes et al. (2020a) does not estimate emissions. | Observation on Gomes et al. (2020a), Table 2, changed to "The study estimates the amount of carbon released in combustion, used as a proxy for estimates of fire-associated emissions." |
| Line 579: Please remove "and even in the units". Gomes et al. (2020a) does not estimate emissions but rather the amount of carbon released in combustion, which is a parameter that can and will be used to estimate emissions. | Sentence removed and now reads: "The difference in values (Table 2) indicates"... |
| Lines 579-584 and 592-594: Seems Discussion to me. | Lines 579-582: We appreciate your comment. Although we agree that this paragraph fit the Discussion, we have kept it there because we think it provides a synthesis of Table 2, placed right before this paragraph. From our perspective, moving it to Discussion would hinder the flow of this analysis. Lines 592-594: moved to Discussion (lines 686-687). |
| Line 602: Define IFM. | Integrated Fire Management (IFM) defined, and acronym used in further mentions of the term. |
| Line 619: "small-scale" | Corrected. |
| Line 620: This section of the sentence is confusing: "fire regime characteristics of fire management activities". Maybe "(…) studies that estimate activities associated with fire management activities, such as prescribed | "Fire regime characteristics of fire management activities" replaced by "activities associated with fire management" |

| | |
|---|---|
| burning (…)". | |
| Lines 627-629: The results of Santos et al. (2021) are valid for which region? Better to mention. | Location clarified in the sentence: "Santos et al. (2021) also documents reduced burned area in the late dry season due to fire management **in two Indigenous Territories in the Cerrado**, which led to reduced fire intensity and reduced extreme wildfires, indicating a reduction in further fire emissions." |
| Lines 630-636: Please highlight that this is a result. That is, that this is what the papers cited discuss and inform. | Sentence included in the beginning of the paragraph: "The literature reviewed shows that"... |
| Lines 640-642: The authors mention a "broad and holistic understanding of the role of these emissions in the carbon budget on regional, national and global scales" however, up to this section of the manuscript, there has been no such discussion. As per my comment for Table 2, I suggest writing a paragraph on this. | Thank you for your comment. When we refer to a "broad and holistic understanding," we are referencing the multiple aspects identified in the literature that influence emission estimates—such as fire policy, fire management practices, and fire dynamic parameters—as well as the diverse methodological approaches used to estimate these emissions, as discussed in the earlier sections of the paper. We have made this clearer in the paragraph, which now reads:

"To our knowledge, and according to our search criteria, this is the first systematic literature review to provide an overview of fire emissions in the Cerrado. By analyzing existing literature on fire emissions in the Cerrado, we identified key topics that contribute to a broad and holistic understanding of the role of these emissions in the carbon budget on regional, national and global scales. This understanding includes not only direct fire-related carbon emission, but also the underlying fire dynamic parameters, fire management practices, and fire policies, along with the various methodological approaches used to estimate these." |
| Line 658: I wouldn't say 31% of papers is "most". | "most" replaced by "many" |
| Lines 673-674: Slightly misleading sentence. | This sentence refers to Gomes et al. (2024) |

| | |
|---|---|
| Two papers were identified that estimate emissions in Cerrado. Even if only one goes back to 1985. | being the only paper found in the literature that evaluates net fire emissions. This was made clearer in the sentence: "This literature review identified one study that includes the removal of $CO_2$ by regrowth in the Cerrado, quantifying the net $CO_2$ emissions from the Cerrado fires from 1985–2020 (Gomes et al., 2024)." |
| Line 693: First time that pyrogenic carbon is mentioned in the text. If the authors want to use this term, I suggest using it earlier in the manuscript. | Term removed from sentence. |
| Line 714: Use the acronym previously defined. | "Integrated Fire Management" replaced by its acronym. |
| Line 724: I would also add to the discussing the findings of Andela et al. (2017; A human-driven decline in global burned area), where they found that a recent decrease in savanna fires worldwide is driven by human occupation. | Sentence added to Discussion (lines 706-707): "In fact, Andela et al. (2017) found a decreasing trend in fire activity driven by human activities worldwide" |
| Table 3: Shouldn't FRP be added to the column on the left? | Fire intensity added to Table 3. |
| I would suggest reinforcing the take-away messages in the Conclusion as well (similarly to the Abstract). | We have added two sentences in the Conclusion to reinforce the take-away messages:

"From our literature review process, we found that research on fire emissions in the Cerrado is still overlooked when compared to other savanna ecosystems."

"Thus, this review demonstrates that understanding the placement of fire emissions in the global carbon budget requires a holistic approach that draws together disciplines across fire science, especially in a distinct environment such as the Cerrado, **while reinforcing the urgent need for further investigation into the topic.**" |